# VAB-8/KIF26, LIN-17/Frizzled, and EFN-4/Ephrin control distinct stages of posterior neuroblast migration downstream of the MAB-5/Hox transcription factor in *Caenorhabditis elegans*

**Vedant D. Jain, Erik A. Lundquist** [ID]*

The University of Kansas, Program in Molecular, Cellular, and Developmental Biology, KU Center for Genomics, Lawrence, Kansas, United States of America

* erikl@ku.edu

## Abstract

Hox transcription factors are involved in neuronal and neural crest development and differentiation, including migration, but the genetic programs employed by Hox genes to regulate terminal differentiation remain to be defined. In *C. elegans*, the Antennapedia-like Hox factor MAB-5 is both necessary and sufficient to induce posterior migration of the Q lineage neuroblasts and neurons downstream of canonical Wnt signaling. Q lineage fluorescence-activated cell sorting and RNA seq in *mab-5* loss-of-function and gain-of-function backgrounds revealed genes with expression in the Q lineage dependent upon MAB-5. Here, the roles of three *mab-5*-regulated genes in QL lineage posterior migration are delineated, *vab-8*/KIF26, *lin-17*/Fz, and *efn-4*/Ephrin. Live, time-lapse imaging of QL.a and QL.ap posterior migration revealed that this migration occurs in three distinct stages: QL.a migration posterior to QL.p (1st stage); after QL.a division, posterior migration of QL.ap to a region immediately anterior to the anus (2nd stage); and final migration of QL.ap posterior to the final position where it differentiates into the PQR neuron (3rd stage). *vab-8* affected each of the three stages, *lin-17* affected stages two and three, and *efn-4* was required for the third stage of posterior QL.ap migration. Thus, different MAB-5-regulated genes control distinct stages of posterior migration. *mab-20*/Semaphorin, a known interaction partner with *efn-4*, also affected only the third stage similar to *efn-4*. Suppression of *mab-5* gof posterior migration confirmed that these genes act downstream of *mab-5* in posterior migration. Possibly, VAB-8/KIF26 helps deliver distinct molecules to the plasma membrane that mediate distinct stages of migration, including LIN-17/Fz and EFN-4. Surprisingly, failure of stages two and three led to the premature extension of a posterior dendritic protrusion, which normally forms after QL.ap had migrated to its final position and PQR differentiation begins. This suggests a link between migration and differentiation, where differentiation is delayed while migration proceeds. In sum,

**Data availability statement:** All relevant data are within the manuscript and its Supporting Information files.

**Funding:** This work was supported by The National Institutes of Health (NS115467 and GM145499 to EAL). The funder had no role in design, data collection and analysis, decision to publish, or preparation of the manuscript.

**Competing interests:** The authors have declared that no competing interests exist.

this work delineates a transcriptional program downstream of *mab-5*/*Hox* that controls posterior neuroblast migration, in response to Wnt signaling.

## Author summary

The migration of neurons in the developing nervous system is key to normal development, and is perturbed in many neurodevelopmental disorders. In this work using the model organism nematode worm *Caenprhabditis elegans*, a novel genetic network controlling neuronal migration in is described. This network involves the conserved Hox factor MAB-5, which regulates differential gene expression, and genes regulated by MAB-5 to drive posterior neuronal migration. Three distinct phases of posterior migration are described, each regulated discretely by genes regulated by MAB-5. Each gene is conserved in mammals. The kinesin-like molecule VAB-8/KIF26 is required for each phase; the LIN-17/Fz receptor is required for phases 2 and three; and the secreted EFN-4/Ephrin is required only for the final phase 3. This work establishes a novel paradigm for studying neuronal migration in *Caenorhabditis elegans*, and defines a novel genetic cassette downstream of a key, conserved regulator of gene expression in neuronal development.

## Introduction

The migration of neurons and neural crest cells is pivotal in the development and organization of the nervous system. Hox genes have been broadly implicated in nervous system development, including segmental specification, compartmentalization, axon guidance, and cell migration (reviewed by [1]). In *C. elegans*, Hox genes have been implicated in a wide variety of terminal neuronal differentiation events [2–4]. In migration of the Q neuroblasts in *C. elegans*, the MAB-5/Antennapedia Hox factor is both necessary and sufficient for posterior migration but does not affect deeper aspects of neuronal fate or differentiation [5–15]. The bilateral Q neuroblasts undergo identical patterns of division, migration, and neuronal differentiation, but QL on the left migrates posteriorly and QR on the right migrates anteriorly [14,16–19]. Initial migration of QL and QR is independent of Wnt signaling and is regulated by receptors UNC-40/DCC and PTP-3/LAR, which are active in QL but not QR, leading to posterior QL versus anterior QR migrations [20,21]. The second stage of Q migration is Wnt-dependent and relies on EGL-20/Wnt activation of expression of the *mab-5*/*Hox* gene in QL lineage but not QR lineage via canonical Wnt signaling in QL [5–14].

As a determinant of posterior migration, MAB-5 is necessary and sufficient for posterior migration of QL descendants, which migrate anteriorly in *mab-5* loss-of-function (*lof*) mutants. QR descendants migrate posteriorly in *mab-5* gain-of-function (*gof*) conditions [14,22]. MAB-5 has a dual role in the posterior migration of QL descendants [14]. MAB-5 first inhibits the anterior migration of the QL

descendants. It does so by inducing expression of CWN-2/Wnt in QL descendants, which acts with EGL-20/Wnt in an autocrine, non-canonical manner to inhibit anterior migration [15]. While anterior migration is inhibited, MAB-5 reprograms QL descendants to migrate to the posterior [14,15].

Hox "realizator" genes, or gene programs downstream of Hox genes to control differentiation, remain incompletely defined. To identify genes regulated by MAB-5 in the Q lineage, fluorescence-activated cell sorting of Q lineage cells and RNA seq were employed in *wild-type, mab-5* loss-of-function (lof), and *mab-5* gain-of-function conditions [15]. *cwn-2*/*Wnt* was downregulated in *mab-5* lof and upregulated in *mab-5* gof, and was shown to be required downstream of MAB-5 to inhibit anterior migration of QL descendants [15].

Among other genes with reduced Q lineage expression in *mab-5* lof were the atypical, conserved Kinesin *vab-8*/*KIF26, lin-17*/*Fz,* and *efn-4*/*Ephrin* [15]. *vab-8* expression was also increased in *mab-5* gof. *vab-8* has previously been implicated in anterior-posterior cell migration and axon guidance [23–27], *lin-17* encodes a Frizzled Wnt receptor involved in canonical Wnt signaling to activate *mab-5* expression in QL and other cell polarity and neural morphogenesis events [28–34], and *efn-4* encodes an Ephrin secreted guidance molecule involved in axon guidance and cellular repulsion, including acting with MAB-5 in prevention of male tail ray cell fusion [35–39]. *vab-8, lin-17,* and *efn-4* mutants each showed defects in the posterior migration of the QL descendant neuron PQR. Furthermore, each suppressed *mab-5* gof, indicating that they were required downstream of MAB-5 to drive posterior migration. Transgenic expression of *vab-8* and *efn-4* in the Q lineage rescued PQR migration defects, suggesting that they act cell-autonomously in the Q cells for posterior migration. Combined with the RNA-seq expression analyses, these genetic results indicate that *vab-8, lin-17,* and *efn-4* are regulated by MAB-5/Hox in the Q lineage that are required for MAB-5-dependent posterior migration.

To understand the cellular nature of these mutant effects on PQR migration, live, time-lapse imaging of QL lineage development was conducted. It was shown previously that in wild-type, after QL division to form QL.a and QL.p (the anterior and posterior daughters), QL.a migrates posteriorly over QL.p to reside in a position posterior to QL.p [14]. In this work, further migration of QL.a was analyzed. After migration to the posterior of QL.p, QL.a underwent division to form QL.aa and QL.ap. QL.aa undergoes programmed cell death, and QL.ap will differentiate into PQR. After division, QL.ap extended a posterior lamellipodial protrusion and migrated posteriorly to a position just anterior to the anus. Here, QL.ap paused for approximately one hour and extended a lamellipodial protrusion to the posterior across into the WT location. QL.ap then followed this lamellipodial protrusion and migrated posteriorly into the WT location. At this point, QL.ap reaches its final position and begins differentiation into the PQR neuron, as evidenced by the extension of a posterior dendritic protusion tipped with a growth cone, which will form the posterior ciliated dendrite of the PQR neuron. This work defined three stages of QL.a and QL.ap migration: the first stage was the migration of QL.a to the posterior of QL.p, where it divided; the second stage was migration of QL.ap posteriorly to a region immediately anterior to the anus; and the third stage migration of QL.ap posterior into the WT location, where it began differentiation into PQR.

*vab-8* mutants displayed failures of each of the three stages of migration, *lin-17* displayed failures of the second and third stages, and *efn-4* displayed failure of only the third stage. This indicated that the three stages were independently controlled by distinct programs downstream of MAB-5. Suppression of *mab-5* gof indicated that *lin-17* acts both upstream of *mab-5* in canonical Wnt signaling, and downstream of *mab-5* in posterior migration. *mab-20*/*Semaphorin,* which acts with *efn-4* in the prevention of male tail ray cell fusion, showed an identical phenotype to *efn-4,* suggesting that their interaction is conserved in posterior QL lineage migration. *mab-20* expression was not affected by MAB-5, suggesting that MAB-20 is a constitutively expressed cofactor that acts with EFN-4 in posterior migration. Double mutant analysis and transgenic rescue indicated that these genes might all act in a common pathway in posterior migration.

A surprising finding was that upon migration failures of the second and third stages, QL.ap immediately extended a posterior dendritic process, one to two hours earlier than *wild-type*. This suggests a balance of migration and differentiation in QL.ap, such that when migration fails, differentiation begins. In short, differentiation of a neuron is actively being suppressed by its migratory program.

In sum, this work has defined a three-step posterior migration process of QL.a and QL.ap controlled by MAB-5/Hox. It delineates a developmental pathway controlling cell migration from a Wnt ligand and canonical Wnt signaling, to a terminal differentiation factor, to molecules involved in interactions of the cell with the surrounding extracellular environment (*i.e.,* novel Hox realizator genes). MAB-5 drives the expression of VAB-8/KIF26, which mediates each of the three stages. MAB-5 might maintain the expression of LIN17/Fz in the QL lineage, which mediates the second and third stages. Finally, MAB-5 drives expression of EFN-4/Ephrin, which is key to the third stage. Given the molecular functions of these molecules, a tempting model is that distinct receptors or receptor protein complexes mediate each step by modifying QL.a or QL.ap interaction with the extracellular environment, and that VAB-8 is required to help deliver these receptors or complexes to the cell surface at each step, mediating the function of MAB-5/Hox as a terminal differentiation factor in posterior QL lineage development.

## Results

### MAB-5/Hox regulates the expression of *vab-8, lin-17/Fz,* and *efn-4/Ephrin* in the QL neuroblast lineage

Previously, Q neuroblasts at the time of *mab-5* expression in QL were sorted by fluorescence-activated cell sorting (FACS) and subjected to RNA-seq [15]. Q cell transcriptomes from *wild-type*, two *mab-5* loss-of-function (lof) mutants (*gk670* and *e1239*) and two *mab-5* gain-of-function (gof) conditions (*e1751* and *lqIs221*) were analyzed for differential expression [15]. Among the genes with expression significantly affected by *mab*-5 after correction for multiple testing were *vab-8, efn-4,* and *lin-17. vab-8* expression was significantly reduced in both *lof* mutants and significantly increased in both *gof* conditions (Table 1). *lin-17/Fz* and *efn-4/Ephrin* were significantly reduced in the *gk670* lof mutant but not significantly affected in other *lof* or *gof* conditions (Table 1). *efn-4* was reduced with *p* value significance in *e1239* lof ($p=0.00904969$), but this did not survive correction for multiple testing ($q=0.2812659$). All three genes were significantly enriched in Q cells compared to whole-animal (Table 1). These results indicate that *vab-8, lin-17,* and *efn-4* are expressed in the Q cells and that their expression might require MAB-5/Hox function, presumably in the QL lineage, although this cannot be established from these studies. *vab-8* showed a paired response (down in lof, up in gof), but *efn-4* and *lin-17* were only affected in lof conditions. Inconsistency of expression changes across genotypes could indicate that these genes are not regulated by MAB-5. However, variability in the FACS sorting and bulk RNA-seq approach could also explain inconsistencies. For example, some genes might be expressed at specific times or in specific cells, and slight variations in the timing of Q cell isolation in biological replicates could limit the ability to identify these genes consistently.

modENCODE MAB-5 CHiP-seq data on Wormbase from embryos, L2, and L3 larvae [40,41] were analyzed for potential MAB-5 binding sites. Significant MAB-5 ChIP-seq peaks were present near *vab-8* in embryos and L2 larvae. *lin-17* also showed nearby significant peaks in embryos, L2, and L3 larval samples. *efn-4* showed no significant peaks. These data suggest that MAB-5 might act at the promoters of *vab-8* and *lin-17* but not *efn-4*. However, Q migration occurs in the L1 larva, which was not represented in this data. Any transient or cell-specific interactions of MAB-5 with the *efn-4* region might not be apparent in these data.

### *vab-8* and *efn-4/Ephrin* control posterior migration of the QL descendant PQR neuron

The PQR neuron is a descendant of the QL neuroblast (PQR is the QL.ap cell in the QL lineage) (Fig 1A) [17]. After QL divides, the anterior daughter QL.a migrates posteriorly over the posterior daughter QL.p and then divides [14,18]. The anterior daughter QL.aa undergoes programmed cell death. The posterior daughter, QL.ap, migrates posteriorly into the tail to reside in the region of the phasmid ganglia in the tail and differentiates into the PQR neuron (Fig 1A-C). The QL.p descendants do not migrate and differentiate into PVM and SDQL near the place of birth (Fig 1A and B). *mab-5* is required for PQR posterior migration. In *mab-5* mutants, PQR migrates anteriorly, similar to AQR [14,42]. AQR is the equivalent neuron from the QR lineage (QR.ap) (Fig 1A and B) and migrates anteriorly to reside near the anterior deirid ganglion in

**Table 1. *mab-5*-dependent Q cell expression of candidate genes.**

| gk670 lof vs wild type | Log2(FC) | p-value | P-Adj (q-value) | |
|---|---|---|---|---|
| efn-4 | -1.3187655 | 2.49E-06 | 0.0007293 | * |
| vab-8 | -1.5960205 | 1.43E-08 | 1.31E-05 | * |
| lin-17 | -1.1287251 | 6.62E-04 | 3.62E-02 | * |
| mab-20 | -0.2976426 | 0.44559032 | 0.80373368 | ns |
| **e1239 lof vs wild type** | | | | |
| efn-4 | -1.0840528 | 0.00904969 | 0.2812659 | ns |
| vab-8 | -1.957639 | 1.01E-07 | 0.0001517 | * |
| lin-17 | -0.58567531 | 0.175745343 | 0.7899871 | ns |
| mab-20 | -0.7631037 | 0.09780806 | 0.67518559 | ns |
| **e1751 gof vs wild type** | | | | |
| efn-4 | -0.0356306 | 0.91395471 | 0.9749122 | ns |
| vab-8 | 1.06637539 | 0.00105495 | 0.0420146 | * |
| lin-17 | 0.156495231 | 0.659561038 | 0.8889923 | ns |
| mab-20 | -0.4109351 | 0.34070904 | 0.71924823 | ns |
| **lqIs221 gof vs wild type** | | | | |
| efn-4 | 0.11188834 | 0.73522122 | 0.9790178 | ns |
| vab-8 | 1.43153159 | 3.31E-05 | 0.004217 | * |
| lin-17 | -0.35149935 | 0.35679139 | 0.8991374 | ns |
| mab-20 | -0.54631 | 0.21602263 | 0.81102639 | ns |
| **Enrichment in Q cells** | | | | |
| efn-4 | 1.99286731 | 2.55E-07 | 4.25E-06 | * |
| vab-8 | 1.77179746 | 0.00019876 | 0.0001517 | * |
| lin-17 | 2.994018727 | 2.61E-08 | 5.77E-07 | * |
| mab-20 | 2.72859827 | 0.00023302 | 0.00151461 | * |

*q<0.05 Data taken from Paolillo et al., 2024.

the head (Fig 1A-C). The QR.p descendants also migrate anteriorly and differentiate into AVM and SDQR neurons (Fig 1A and 1B).

*vab-8* was previously reported to affect PQR migration [23]. The positions of AQR and PQR were determined in *vab-8, lin-17,* and *efn-4* mutants (see Materials and Methods). AQR migration was unaffected in these mutants (Table 2). In contrast, PQR displayed significant failure of posterior migration in *vab-8, lin-17,* and *efn-4* (Table 2 and Fig 1D and 1E). In *vab-8(ev411)* and *vab-8(e1017)*, 52% and 62% of PQR failed to migrate to the normal position in the tail. *vab-8(e1017)* causes a premature stop codon predicted to affect all *vab-8* isoforms A, B and C and is likely a null [24,27]. *vab-8(ev411)* alters the splice donor sequence of intron three and is predicted to affect only the *vab-8A* long isoform [26], indicating that the long isoform of *vab-8* is required for PQR migration. In a screen for mutants with defective PQR migration, two new *vab-8* alleles were isolated. *vab-8(lq153)* and *vab-8(lq156)* each affect PQR migration with no effects on AQR (Table 2). *vab-8(lq153)* affected isoforms A and B and was a C to T mutation that resulted in R127 to stop in *vab-8C*. *vab-8(lq156)* was a C to T mutation in the *vab-8A* long isoform that resulted in a Q46 to stop. These new *vab-8* mutations confirm the role of *vab-8* in PQR migration.

In *efn-4(bx80)* and *efn-4(e36)*, 71% and 81% failed to migrate posteriorly. A small number of PQR neurons migrated anteriorly in *vab-8* mutants (0–6%, positions 1–3), but the vast majority failed to migrate posteriorly (position 4) (Table 2). This suggests that *vab-8* and *efn-4* primarily affect the ability of PQR to migrate to the posterior.

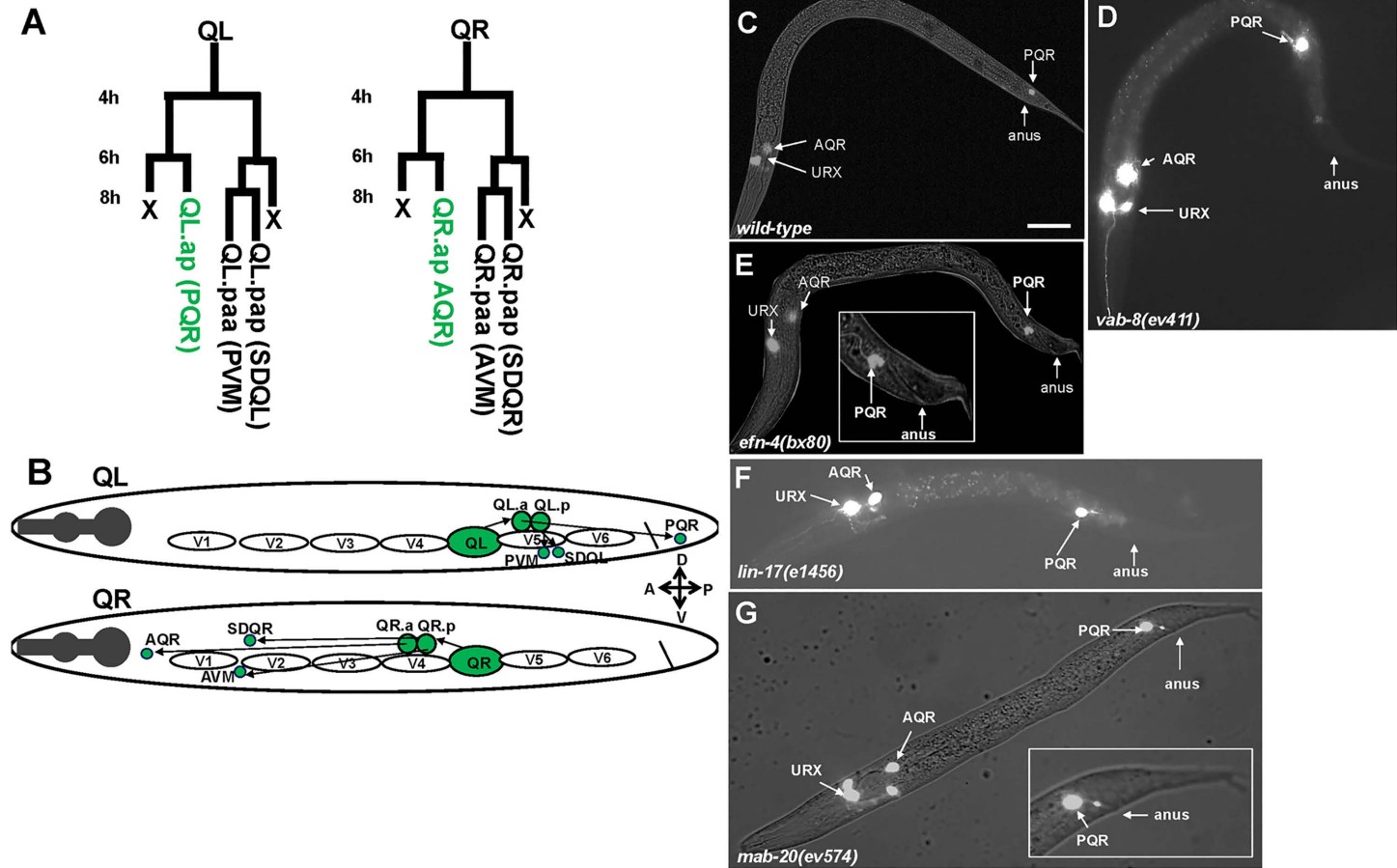

**Fig 1. Q neuroblast migration and migration defects. A)** The Q neuroblast lineage in the L1 larva. Hours represent time after hatching. **B)** Schematic of migration of the QL and QR lineages. QL on the left migrates over the V5 seam cell in the first stage of migration and divides into QL.a and QL.p. In the second stage, QL.a continues posterior migration and divides into QL.aa and QL.ap. QL.ap migrates into the tail of the animal and differentiates into the PQR neuron. QL.p does not migrate and differentiates into the PVM and SDQL neurons. On the right, QR migrates anteriorly over V4 and divides. Both QR.a and QR.p migrate anteriorly and differentiate into AQR, AVM, and SDQR. **C-G)** Composite fluorescent/DIC micrographs showing the position of AQR, PQR, and URX neurons in different genotypes using the *lqIs244[Pgcy-32::gfp]* transgene. The scale bar in A represents 10μm. **C)** A wild type animal with PQR in the WT location (position 5 in Table 2) and AQR near the anterior deirid (position 1 in Table 2). **D)** A *vab-8(ev411)* animal with a PQR that failed to migrate anteriorly (position 4 in Table 2; paradigm 1 in Table 4). **E)** An *efn-4(bx80)* animal with a PQR that failed to complete full migration, residing immediately anterior to the anus (paradigm 2 in Table 4). The inset is a magnified view of the tail region. **F)** A *lin-17(e1456)* mutant with a failed posterior PQR migration (paradigm 1 in Table 4). **G)** A *mab-20(ev574)* mutant with a failed PQR migration immediately anterior to the anus (paradigm 2 in Table 4).

*efn-4(bx80)* is a deletion of exon 2 [37,43], and *efn-4(e36)* is a premature stop codon in exon 2 [36]. Both are predicted to be null for *efn-4.*

### lin-17/Fz might act both upstream and downstream of *mab-5* in PQR migration

*lin-17* has been shown previously to affect PQR migration [14,42]. *lin-17(e1456), lin-17(n671),* and *lin-17(lq202)* displayed 62%, 70%, and 62% defective PQR migration (Table 2 and Fig 1F), with most defective PQR failing to migrate to the posterior (position 4). However, each also showed significant anterior migration of PQR (8% for *e1456,* 23% for *n671,* and 32% for *lq202*). *lin-17* acts in canonical Wnt signaling that activates *mab-5* expression in QL [7,12,44,45]. The anterior

**Table 2. AQR and PQR migration defects in mutants.**

| Genotype[1] | AQR[2] | | | | | PQR[2] | | | | |
|---|---|---|---|---|---|---|---|---|---|---|
| | 1 | 2 | 3 | 4 | 5 | 1 | 2 | 3 | 4 | 5 |
| *Candidate Screens* | | | | | | | | | | |
| *wild-type* | 100 | 0 | 0 | 0 | 0 | 0 | 0 | 0 | 0 | 100 |
| *efn-4(bx80)* | 100 | 0 | 0 | 0 | 0 | 0 | 0 | 1 | 70 | 29 |
| *efn-4(e36)* | 100 | 0 | 0 | 0 | 0 | 0 | 0 | 0 | 81 | 19 |
| *vab-1(e2)* | 100 | 0 | 0 | 0 | 0 | 0 | 0 | 0 | 0 | 100 |
| *vab-8(ev411)* | 100 | 0 | 0 | 0 | 0 | 1 | 0 | 1 | 50 | 48 |
| *vab-8(e1017)* | 100 | 0 | 0 | 0 | 0 | 1 | 1 | 5 | 55 | 38 |
| *vab-8(lq153)* | 100 | 0 | 0 | 0 | 0 | 0 | 0 | 0 | 70 | 30 |
| *vab-8(lq156)* | 100 | 0 | 0 | 0 | 0 | 0 | 0 | 0 | 56 | 44 |
| *lin-17(e1456)* | 100 | 0 | 0 | 0 | 0 | 1 | 3 | 4 | 54 | 38 |
| *lin-17(n671)* | 100 | 0 | 0 | 0 | 0 | 10 | 4 | 8 | 48 | 30 |
| *lin-17(lq202)* | 100 | 0 | 0 | 0 | 0 | 16 | 9 | 7 | 36 | 32 |
| *mab-20(ev574)* | 100 | 0 | 0 | 0 | 0 | 2 | 0 | 1 | 50 | 47 |
| *mab-20(bx24)* | 100 | 0 | 0 | 0 | 0 | 0 | 0 | 0 | 55 | 45 |
| *mab-20(bx61)* | 100 | 0 | 0 | 0 | 0 | 0 | 0 | 0 | 4 | 96 |
| *mab-20(ev778)* | 100 | 0 | 0 | 0 | 0 | 0 | 0 | 0 | 36 | 64 |
| *mom-2(or77)M+* | 100 | 0 | 0 | 0 | 0 | 0 | 0 | 0 | 19 | 81 |
| *egl-20(mu39)* | 100 | 0 | 0 | 0 | 0 | 29 | 16 | 9 | 8 | 38 |
| *cwn-2(ok895)* | 100 | 0 | 0 | 0 | 0 | 0 | 0 | 0 | 9 | 91 |
| *cwn-1(ok546)* | 100 | 0 | 0 | 0 | 0 | 0 | 0 | 0 | 1 | 99 |
| *lin-44(n1792)* | 100 | 0 | 0 | 0 | 0 | 0 | 0 | 0 | 0 | 100 |
| *Suppression experiments* | | | | | | | | | | |
| *Pegl-17::mab-5* | 0 | 0 | 0 | 5 | 95 | 0 | 0 | 0 | 0 | 100 |
| *mab-5(e1751)* | 1 | 1 | 0 | 16 | 82 | 0 | 0 | 0 | 1 | 99 |
| *efn-4(bx80); Pegl-17::mab-5* | 0 | 0 | 0 | 81* | 19 | 0 | 0 | 0 | 66 | 34 |
| *vab-8(ev411); Pegl-17::mab-5* | 2 | 2 | 7 | 72* | 17 | 0 | 0 | 1 | 72 | 27* |
| *vab-8(e1017); Pegl-17::mab-5* | 0 | 1 | 9 | 75* | 15 | 0 | 0 | 4 | 79 | 17* |
| *lin-17(e1456); Pegl-17::mab-5* | 0 | 0 | 0 | 75* | 25 | 0 | 0 | 0 | 72 | 28* |
| *lin-17(n671); Pegl-17::mab-5* | 0 | 0 | 0 | 85* | 15 | 0 | 0 | 0 | 90 | 10* |
| *lin-17(lq202); Pegl-17::mab-5* | 0 | 0 | 0 | 68* | 32 | 0 | 0 | 0 | 69 | 31 |
| *mab-20(ev574); Pegl-17::mab-5* | 0 | 0 | 0 | 67* | 33 | 0 | 0 | 0 | 68 | 32* |
| *mab-20(ev574); mab-5(e1751)* | 0 | 0 | 0 | 72* | 28 | 0 | 0 | 0 | 72 | 28* |
| *Rescue experiments* | | | | | | | | | | |
| *Pegl-17::efn-4[3]* | 100 | 0 | 0 | 0 | 0 | 0 | 0 | 0 | 0 | 100 |
| *efn-4(bx80); Pegl-17::efn-4[3,4]* | 100 | 0 | 0 | 0 | 0 | 0 | 0 | 0 | 50 | 50* |
| *Pegl-17::vab-8A[3]* | 100 | 0 | 0 | 0 | 0 | 0 | 0 | 0 | 0 | 100 |
| *vab-8(ev411); Pegl-17::vab-8A[3,4]* | 100 | 0 | 0 | 0 | 0 | 0 | 0 | 0 | 35 | 65* |
| *Pegl-17::vab-8C[5]* | 100 | 0 | 0 | 0 | 0 | 0 | 0 | 0 | 0 | 100 |
| *vab-8(e1017); Pegl-17::vab-8C[4,5]* | 100 | 0 | 0 | 0 | 0 | 0 | 0 | 0 | 46 | 54* |
| *Pegl-17::mab-20[3]* | 100 | 0 | 0 | 0 | 0 | 0 | 0 | 0 | 0 | 100 |
| *mab-20(ev574); Pegl-17::mab-20[3,4]* | 100 | 0 | 0 | 0 | 0 | 0 | 0 | 0 | 27 | 73* |
| *Double mutants* | | | | | | | | | | |
| *efn-4(bx80); vab-8(ev411)* | 100 | 0 | 0 | 0 | 0 | 0 | 0 | 11 | 80 | 9 |

*(Continued)*

**Table 2.** (Continued)

| Genotype[1] | AQR[2] | | | | | PQR[2] | | | | |
|---|---|---|---|---|---|---|---|---|---|---|
| | 1 | 2 | 3 | 4 | 5 | 1 | 2 | 3 | 4 | 5 |
| *efn-4(bx80); vab-8(e1017)* | 100 | 0 | 0 | 0 | 0 | 1 | 0 | 7 | 87 | 5 |
| *lin-17(e1456); vab-8(ev411)* | 100 | 0 | 0 | 0 | 0 | 4 | 1 | 12 | 79 | 4 |
| *lin-17(n671); vab-8(ev411)* | 100 | 0 | 0 | 0 | 0 | 8 | 4 | 6 | 58 | 24 |
| *lin-17(lq202); vab-8(ev411)* | 100 | 0 | 0 | 0 | 0 | 11 | 10 | 11 | 48 | 20 |
| *lin-17(e1456); efn-4(bx80)* | 100 | 0 | 0 | 0 | 0 | 2 | 0 | 1 | 76 | 21 |
| *lin-17(n671); efn-4(bx80)* | 100 | 0 | 0 | 0 | 0 | 4 | 5 | 5 | 78 | 8 |
| *lin-17(lq202); efn-4(bx80)* | 100 | 0 | 0 | 0 | 0 | 7 | 8 | 10 | 65 | 10 |
| *efn-4(bx80); mab-20(ev574)* | 100 | 0 | 0 | 0 | 0 | 0 | 0 | 0 | 88 | 12 |
| *vab-8(ev411); mab-20(ev574)* | 100 | 0 | 0 | 0 | 0 | 0 | 0 | 0 | 70 | 30 |
| *cwn-2(ok895); mom-2(or77)M+* | 100 | 0 | 0 | 0 | 0 | 0 | 0 | 0 | 33 | 67 |
| *Compensation* | | | | | | | | | | |
| *mab-20(ev574); Pegl-17::efn-4*[4,5] | 100 | 0 | 0 | 0 | 0 | 0 | 0 | 0 | 17 | 83* |
| *efn-4(bx80); Pegl-17::mab-20*[4,5] | 100 | 0 | 0 | 0 | 0 | 0 | 0 | 0 | 36 | 64* |

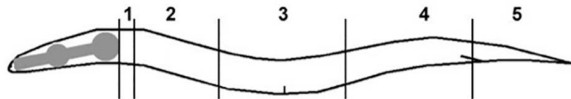

[1] For each genotype, 100 animals were scored on three separate occasions for a total of 300 animals.

[2] Numbers are average percentages from three replicates of cells at each position rounded to the nearest whole number.

[3] Three independent transgenes were scored with similar effects.

[4] Number of WT PQRs compared against the single mutant of the respective candidate.

[5] One transgene was scored.

*$p<0.05$

migration of PQR in *lin-17* might be due to the role of *lin-17* in canonical Wnt signaling upstream of *mab-5* in QL, and the failure of PQR to migrate posteriorly might be due to a distinct role in posterior PQR migration. *mab-5 gof* suppression experiments described below are consistent with this hypothesis.

*lin-17(e1456)* alters the splice acceptor sequence of intron 8 [44], and given its weaker PQR phenotype, might be hypomorphic. *lin-17(n671)* introduces a premature stop codon in exon 9 and is likely null [44]. *lin-17(lq202)* was found as a spontaneous mutation affecting PQR migration and is a 75-bp in-frame deletion in exon 8 that removes a portion of the second transmembrane domain (see Materials and Methods). PQR migration defects resemble those of *lin-17(n671)* and therefore *lin-17(lq202)* is a likely null.

### *vab-8, lin-17,* and *efn-4* lof mutants suppress *mab-5* gof

In *mab-5(e1751) gof* mutants, *mab-5* is ectopically expressed in QR and QL, and the *lqIs220* and *lqIs221* transgenes drive *mab-5* expression in both QL and QR using the *egl-17* promoter [15,22,46]. In each case, both AQR and PQR migrate posteriorly (Table 2 and Fig 2A). Double mutants of *vab-8, lin-17,* and *efn-4* with the *mab-5* duplication allele *mab-5(e1751)* were lethal and could not be constructed.

In *lqIs220[Pegl-17::mab-5]*, 95% of AQRs migrated posteriorly to the tail, similar to PQR, with no AQR migrating anteriorly (Table 2 and Fig 2A). In *vab-8(ev411); Pegl-17::mab-5* and *vab-8(e1017); Pegl-17::mab-5*, 17% and 15% of AQR migrated posteriorly to the tail, compared to 95% in *Pegl-17::mab-5* alone (Table 2 and Fig 2B). The majority of AQRs that failed to migrate posteriorly remained near the birth position (position 4). Similarly, 27% and 17% of PQR migrated

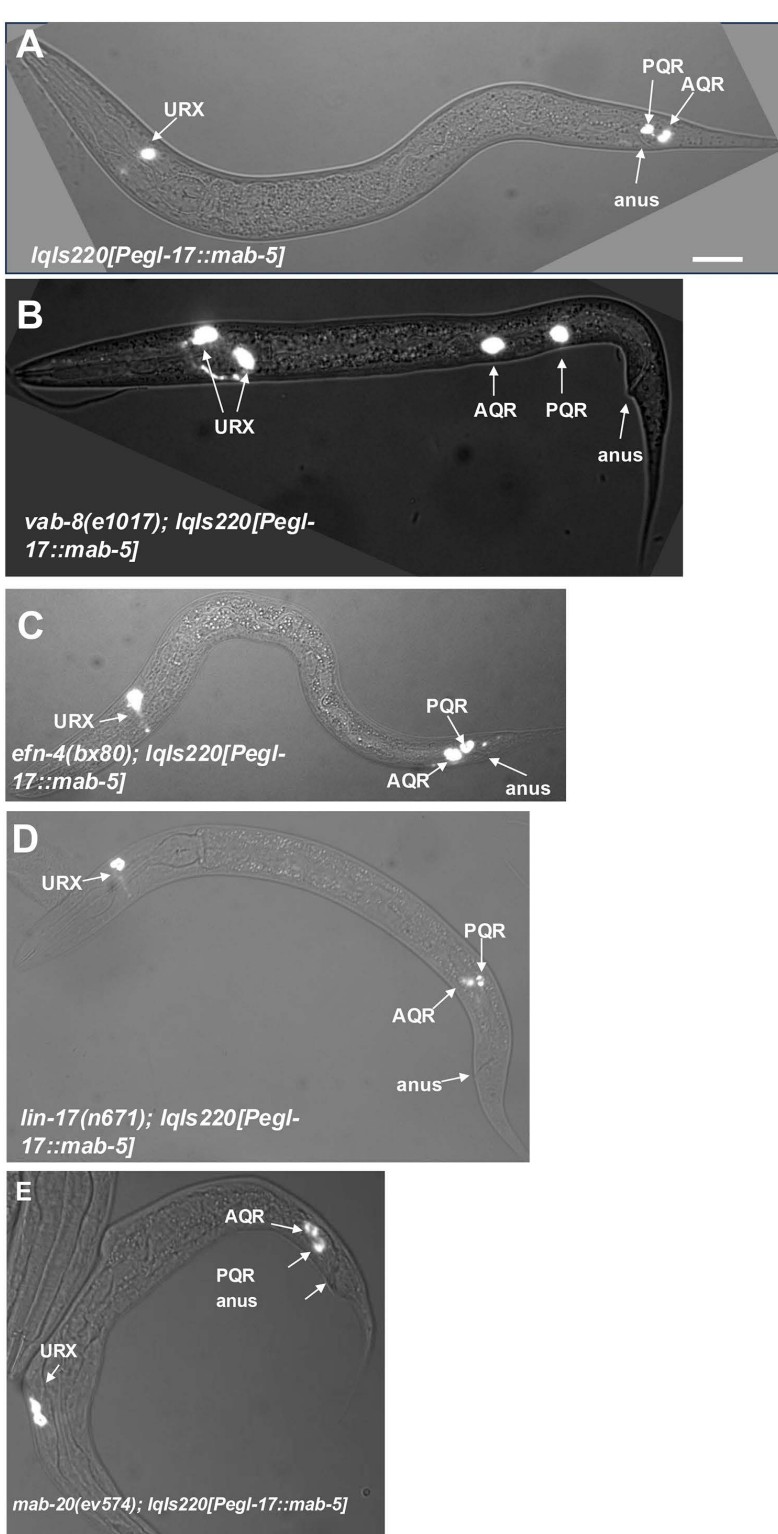

**Fig 2. Suppression of *mab-5* gain-of-function by *vab-8, efn-4,* and *lin-17*. A]** Composite fluorescent/DIC micrographs showing the positions of AQR and PQR as described in Fig 1. The scale bar in A represents 10μm. **A)** An animal harboring the *lqIs220[Pegl-17::mab-5(+)]* transgene that drives *mab-5* expression in both QL and QR lineages. Both AQR and PQR migrated posteriorly in the tail (position 5 in Table 2). **B)** In a *vab-8(e1017); lqIs220* animal,

both AQR and PQR failed to migrate behind the anus and stayed in position 4. **C)** AQR and PQR failed to fully migrate posteriorly in an *efn-4(bx80); lqIs220* mutant and resided immediately anterior to the anus. **D)** AQR and PQR failed to migrate posteriorly in a *lin-17(n671); lqIs220* animal. **E)** AQR and PQR failed to fully migrate to the posterior and resided immediately anterior to the anus in a *mab-20(ev574); lqIs220* animal.

posteriorly in *vab-8(ev411); Pegl-17::mab-5* and *vab-8(e1017); Pegl-17::mab-5*, with the remaining PQR residing at the Q cell birth position 4. These data indicate that *vab-8* is required for ectopic MAB-5 to drive posterior migration of both AQR and PQR and suggest that VAB-8 acts downstream of MAB-5. Notably, some AQR neurons migrated anteriorly in *vab-8; Pegl-17::mab-5* (Table 2). This suggests that *vab-8* might also control the direction of AQR migration downstream of MAB-5. This is consistent with the low level of anterior PQR migration seen in *vab-8* mutants alone (Table 2).

In *vab-8; Pegl-17::mab-5* animals, the percent of PQRs in the wild-type position 5 was significantly lower than the *vab-8* alone (27% and 17% compared to 48% and 38%) (Table 2). The nature of this interaction is not understood, but it is possible that MAB-5 transgenic expression can inhibit PQR posterior migration in the absence of VAB-8.

The *efn-4(bx80); Pegl-17::mab-5* animals showed a significant reduction in the number of AQRs that migrated posteriorly (19% compared to 95% for *lqIs220* alone) (Table 2 and Fig 2C). Cells that failed to migrate posteriorly all remained immediately anterior in position 4, and none migrated anteriorly (Table 2 and Fig 2C). PQR migration defects resembled *efn-4* single mutants (Table 2). Thus, *efn-4* likely acts downstream of *mab-5* specifically in PQR posterior migration.

*lin-17* mutants alone displayed a failure of PQR posterior migration as well as significant anterior PQR migration (Table 2 and Fig 2D). *lin-17; Pegl-17::mab-5* animals displayed no anterior PQR migration (Table 2), suggesting that ectopic *mab-5* rescued *lin-17* anterior PQR migration. This is consistent with the known role of *lin-17* in canonical Wnt signaling that activates *mab-5* expression in QL.

*lin-17; Pegl-17::mab-5* mutants displayed a failure of posterior AQR migration (25% and 33% in the posterior compared to 95% in *Pegl-17::mab-5* alone) (Table 2). Thus, *lin-17* suppressed *mab-5* gof for posterior AQR migration. PQR posterior migration defects were increased in two of the three *lin-17; Pegl-17::mab-5* strains compared to *lin-17* single mutants (28% and 10% compared to 38% and 30%) (Table 2). *lin-17(lq202); Pegl-17::mab-5* did not display significantly increased PQR defects. Similar to *vab-8,* it is possible that transgenic *mab-5* expression can inhibit posterior PQR migration in the absence of *lin-17*. In any event, these results suggest that *lin-17* acts both upstream and downstream of *mab-5*: *lin-17* participates in canonical Wnt signaling upstream of *mab-5* to activate its expression; and *lin-17* acts downstream of *mab-5* to drive posterior migration.

Overall, these results suggest that *vab-8, lin-17,* and *efn-4* are regulated by *mab-5*, and act downstream of *mab-5.* In the absence of these genes, *mab-5* is incapable of fully executing its role as a determinant of posterior migration of PQR.

### *vab-8* and *efn-4* can act cell-autonomously in the Q cells

Transgenes were constructed that drive *vab-8* and *efn-4* cDNA expression in the Q lineages using the *egl-17* promoter (see Materials and Methods). The long isoform *vab-8A* partially but significantly rescued *vab-8(ev411)* (48% to 65% wild-type PQR), and the short isoform *vab-8C* partially but significantly rescued the posterior PQR migration of *vab-8(e1017)* (38% to 54% wild-type PQR) (Table 2). *Pegl-17::efn-4* also partially but significantly rescued *efn-4(bx80)* (29% to 50% wildtype PQR) (Table 2). These results suggest that *vab-8* and *efn-4* can act cell autonomously. However, the rescue was incomplete. This could be due to the transgenic expression of these genes, which would not duplicate any fine-scale timing or levels of expression that might be required for full rescue. It is also possible that they also have non-autonomous roles outside of the Q cells. In any event, *vab-8* and *efn-4* can at least partially act cell-autonomously in the Q cells and their intracellular expression is necessary for migration.

Neither *Pegl-17::vab-8* transgenes or *Pegl-17::efn-4* alone caused any defects in AQR or PQR migration (Table 2). This indicates that while *vab-8* and *efn-4* are required for posterior PQR migration, they are not sufficient to cause posterior AQR migration as is *mab-5*. Possibly, multiple, redundant pathways act downstream of MAB-5 to drive posterior migration, including VAB-8 and EFN-4.

### *In vivo* live imaging uncovers distinct stages of QL.ap (PQR) migration

In the first 4–5 hours of larval development, Q neuroblasts undergo the initial Wnt and MAB-5-independent stage of migration [14,18,42]. QL protrudes and migrates posteriorly above the V5 seam cell, and QR protrudes and migrates anteriorly over the V4 seam cell. Each then undergo its first division to produce QX.a and QX.p. It is at this point that Wnt signaling and MAB-5 control QL.a/p migration [14,18,42]. QL.p does not normally migrate, but QL.a migrates posteriorly over QL.p. The first migration of QL.a has been previously described with live imaging [14,18].

To define QL.ap migration, live imaging of early L1 larvae immobilized in a vivoChip microfluidic device from vivoVerse was conducted (see Materials and Methods). Embryos were isolated and allowed to hatch in M9 media without food overnight, resulting in early L1 arrest. These arrested L1s were placed on food and allowed to develop for 4–5 hours before imaging. Using this protocol, QL.a cells had already migrated posterior to QL.p (Fig 3A). QL.a underwent an asymmetric cell division after ~30–45 minutes to produce QL.aa and QL.ap (Fig 3B). QL.aa underwent programmed cell death. QL.ap extended a lamellipodial protrusion to the posterior and, at ~60 minutes, had migrated to a position just posterior to the anus (Fig 3B and 3C). At this point, QL.ap extended a broad lamellipodia protrusion to the posterior, and remained in this position for ~60 minutes (Fig 3C). At ~120 minutes, the QL.ap cell body migrated posteriorly in the WT location (Fig 3D). After this migration, differentiation into the PQR neuron began, including extension of a posterior dendritic growth cone [33,47] (Fig 1E). All 50 *wild-type* animals imaged displayed this pattern (Table 3). In summary (Fig 3E), posterior migration of QL.ap resembled the initial saltatory migration of the Q neuroblasts [42]: the cell extended a lamellipodial protrusion in the direction of migration, paused, and then somal translocation in the direction of lamellipodial protrusion. The QL.a lineage undergoes three such saltatory migrations (Fig 3F): migration posteriorly over QL.p; QL.ap migration to a position immediately anterior to the anus; and QL.ap migration in the WT location in the tail.

### *vab-8* is necessary for all three stages of QL.ap migration

*vab-8* mutants had incomplete posterior migration of PQR (Table 2 and Fig 1D). Live imaging as described above was used to analyze QL.a lineage migration in *vab-8(ev411)* animals. When imaging began in *vab-8(ev411)*, all QL.a had divided to form QL.aa and QL.ap (Fig 4A). In *wild-type*, this division occurred after imaging began. *vab-8* might affect the timing of QL.a division.

In 25/46 animals, QL.a had not completely migrated posterior to QL.p before division (Table 4 and Fig 4A). Of the 25/46 QL.a that failed to completely migrate over QL.p before dividing, at 60 minutes, QL.ap had not reached the position immediately anterior to the anus after the second stage but was in a more anterior position (Fig 4B). By 120 minutes, 20/25 QL.ap completed stage three of posterior migration to the normal position of QL.ap posterior in the tail (Fig 4A-C and Table 4 (1 and WT)). This suggests that deficits in migration can be compensated in later migration steps. However, 5/25 did not complete full migration and stopped near the beginning point of the second stage (3/5) (Table 3 (1 and 2)), or the third stage (2/5) (Table 3 (1 and 3)). No failures at both the second and third stages were observed. Thus, the absence of *vab-8* might cause premature timing of QL.a division and/or affect the ability of QL.a to complete the first stage of migration before dividing.

In *vab-8* animals in which QL.a divided posterior to QL.p, the second and third stages sometimes failed. 5/46 QL.ap failed to complete only the second stage (Fig 4D and 4E and Table 4), and 12/46 failed to complete only the third stage and resided immediately to the anterior of the anus (Fig 4F and 4G and Table 4). These results suggest that VAB-8 is required for all three stages of QL.a migration (Fig 4H).

### Migration failure resulted in the premature extension of a posterior dendritic process

After QL.ap completes the third and final stage of migration in the WT location, at 120 minutes, it begins to differentiate into the PQR neuron, including posterior extension of a thin dendritic process tipped with a growth cone [33,47] (Fig 1E). In *vab-8(ev411)* mutants with failures of stages two and three of migration, QL.ap immediately extended a thin posterior

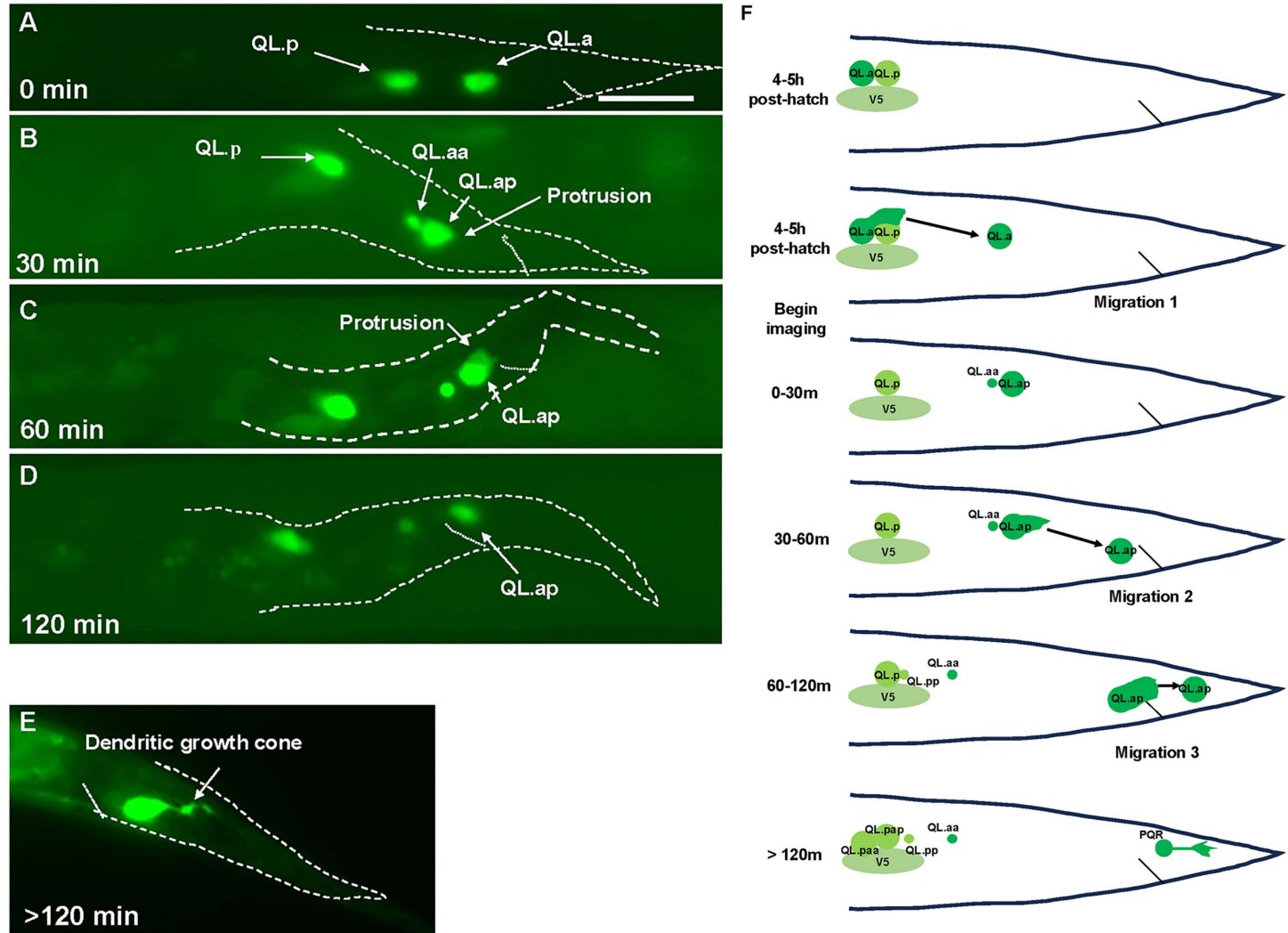

**Fig 3. Time-lapse imaging of QL.a and QL.ap migration in wild-type.** Fluorescent micrographs of animals immobilized in the vivoChip live imaging module from vivoVerse are shown. The animals harbor the *ayIs9[Pegl-17::gfp]* transgene that is expressed in the Q cells and descendants. The scale bar in A represents 10μm. A-D are of the same animal imaged four times over a two-hour period starting at 4-5 hours post-feeding. The times indicate when the images were taken relative to the beginning of the imaging session. **A)** At 4-5 hours post-feeding, QL.a had migrated posteriorly over QL.p (migration 1). **B)** 30 minutes later, QL.a had divided to form QL.aa and QL.ap. QL.aa will undergo programmed cell death, and QL.ap has extended a posterior lamellipodial protrusion. **C)** 60 minutes after imaging began, QL.ap had migrated to a position immediately anterior to the anus, and extended a robust posterior lamellipodial protrusion (migration 2). **D)** Between 60 and 120 minutes after imaging began, QL.ap migrated posterior in the WT location (migration 3). **E)** After QL.ap migration had completed at 120 minutes, it extended a posterior dendritic protrusion as it differentiated into the PQR neuron. This micrograph is of a different animal than those shown in A-D. **F)** A schematic showing the three distinct posterior migrations undertaken by QL.a and QL.ap as it migrates to the final WT location to form the PQR neuron.

dendritic process instead of the broad lamellipodial protrusion of migrating cells (compare Figs 3C and 4G). Failure at stage two resulted in dendritic extension ~120 minutes before *wild-type* (Fig 4D and 4E). In Fig 4D, a dendritic protrusion is seen as soon as imaging begins, and by 60 minutes in Fig 4E, the dendritic growth cone had reached the WT location. In Fig 4F, QL.ap is beginning the second stage of migration 60 minutes after imaging began, and shows the posterior lamellipodial protrusion characteristic of migrating cells. At 120 minutes in Fig 4G, the QL.ap stopped migrating and began

PLOS Genetics

**Table 3. Migration failures in live imaging of animals.**

| Genotype | Migration failure stage | | | | | | | |
|---|---|---|---|---|---|---|---|---|
| | 1 | 2 | 3 | 1 and 2 | 1 and 3 | 1 and WT[1] | No defective stage | N |
| *wild-type* | 0 | 0 | 0 | 0 | 0 | 0 | 50 | 50 |
| *vab-8(ev411)* | 25 | 5 | 12 | 3 | 2 | 20 | 14 | 46 |
| *lin-17(e1456)* | 0 | 2 | 11 | 0 | 0 | 0 | 20 | 33 |
| *efn-4(bx80)* | 0 | 0 | 21 | 0 | 0 | 0 | 19 | 40 |
| *mab-20(ev574)* | 0 | 0 | 14 | 0 | 0 | 0 | 5 | 19 |

[1]"WT" indicates that the QL.ap cell reached the normal wild-type position after having failed the first stage of migration.

extension of a posterior dendritic process ~60 minutes before wild-type. These results suggest that failure to migrate results in premature extension of a posterior dendritic process (Fig 4H), possibly indicating premature differentiation into the PQR neuron.

### lin-17 is required for the second and third stages of migration, but not the first

Similar to *vab-8(ev411)* mutants, QL.a had already divided when imaging began in *lin-17(e1456)* (Fig 5A). However, all QL.a had migrated posteriorly to QL.p to the normal position of the beginning of stage 2 (n = 33) (Table 3 and Fig 5A and C). In *lin-17* mutants, migration defects were noted at the second (2/33) (Fig 5A and 5B and Table 4) and third (11/33) stages of migration (Fig 5C–5E and Table 3). Accompanying migration failure was the premature extension of a posterior dendrite. In Fig 5B, the second stage failed, and QL.ap extended a posterior dendrite at 60 minutes after imaging began. In Fig 5D, QL.ap had completed stage 2 at 60 minutes, but failed in stage three migration and extended a posterior dendritic process in the WT location (Fig 5E). *lin-17(e1456)* displayed stage two and three migration failures (Fig 5F), and, similar to *vab-8(ev411)*, extended posterior dendritic processes prematurely, one to two hours earlier than *wild-type*, depending on where the migration failed.

### efn-4 is necessary for the third and final step of QL.ap migration

Live time-lapse imaging of *efn-4(bx80)* L1 larvae was conducted as described above (n = 40). At the beginning of the imaging process, all 40 QL.a had undergone the posterior migration over QL.p and divided into QL.aa and QL.ap (Fig 6A and Table 4)). However, all QL.a had fully migrated posterior to QL.p before dividing. The second step of migration also occurred normally in all *efn-4(bx80)* mutants (Fig 6B and Table 4). At this point, 21/40 QL.ap failed to extend the broad posterior lamellipodial protrusion seen in *wild-type* and instead, a posterior dendritic process emanated from QL.ap (Fig 6C and Table 4). These QL.ap failed to execute stage three migration posteriorly in the WT location. *efn-4* was required for the third and final migration of QL.ap, and when this migration did not occur, a posterior dendritic process formed prematurely, similar to *vab-8(ev411)* and *lin-17(e1456)* (Fig 6F). This phenotype is consistent with the final position of PQR neurons in *efn-4* at a position immediately anterior to the anus (Fig 1E), whereas *vab-8* and *lin-17* show PQR neurons at this position as well as positions further to the anterior consistent with earlier migration failures. *vab-1* encodes the sole member of the Eph receptor tyrosine kinase in *C. elegans* [48]. However, *vab-1(e2)* mutants displayed no defects in PQR migration (Table 2). Thus, EFN-4 might utilize a distinct receptor in PQR migration.

### The final PQR position reflects the stage of migration failure

The results above indicate three stages of QL.a migration. *vab-8* affects all three, *lin-17* affects stages 2 and 3, and *efn-4* affects only stage 3. Stage 3 failures resulted in QL.ap residing immediately anterior to the anus, as opposed to stage 1 and 2 failures, which resulted in more anterior QL.ap. These positions were reflected in the final positions of PQR in

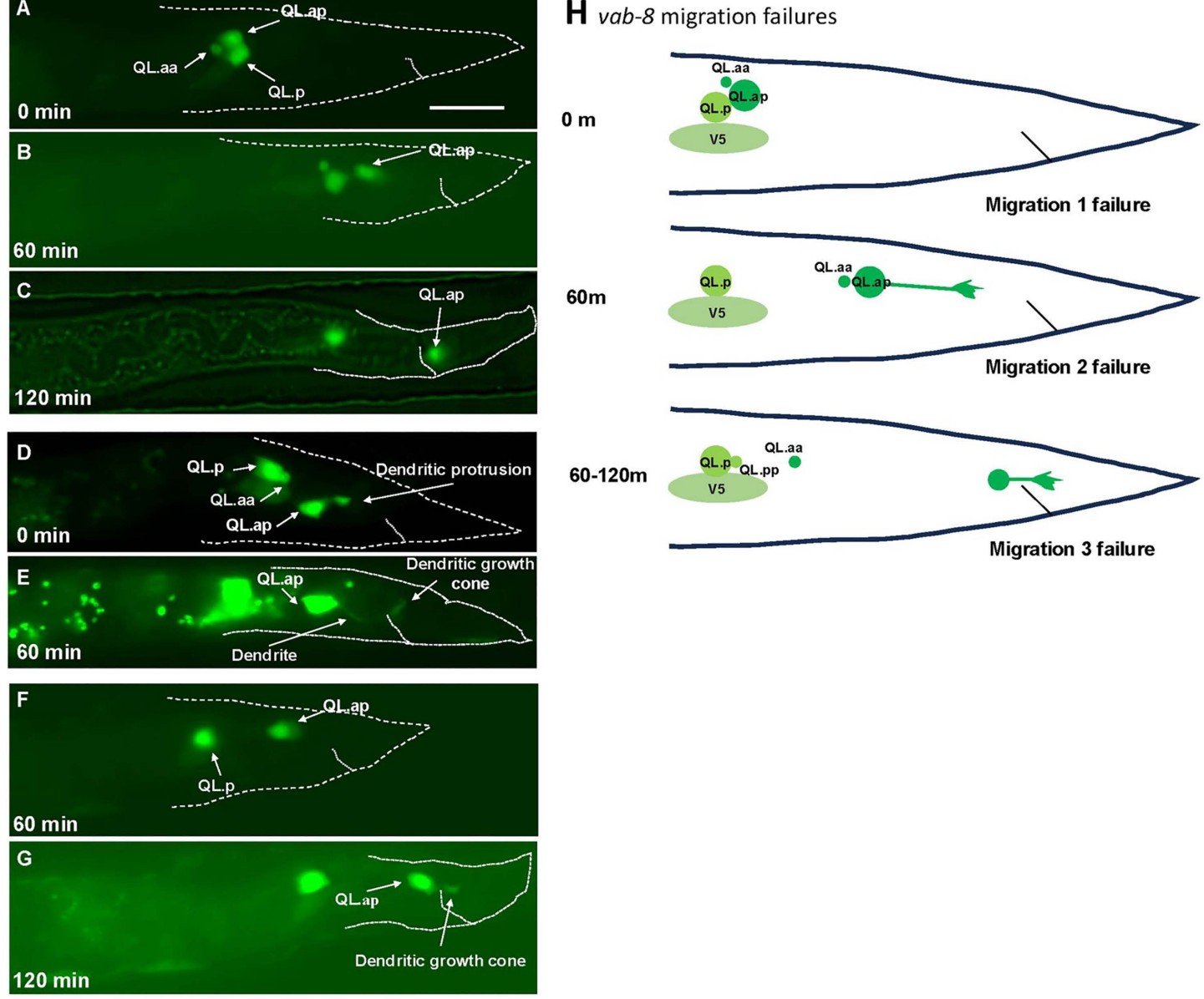

**Fig 4. Time-lapse imaging of QL.a and QL.ap migration in *vab-8(ev411)*.** Fluorescent micrographs of L1 *vab-8(ev411)* animals immobilized and imaged over time lapse as described in Fig 3. **A–C** are the same animal; **D** and **E** are the same animal; and **F** and **G** are the same animal. The scale bar in A represents 10μm. **A)** At 4-5 hours after feeding and at the beginning of imaging, QL.a failed to migrate completely posteriorly to QL.p. QL.a had also divided prematurely, one hour earlier than wild-type. **B)** 60 minutes later, QL.ap executed the second migration. **C)** By 120 minutes after imaging, QL.ap reached the normal position in the WT location despite failing in the first migration over QL.p. **D)** 4-5 hours after feeding when imaging began, QL.a had already migrated posterior to QL.p and divided. A thinner, dendritic protrusion with growth cone emanated posteriorly from QL.ap. **E)** 60 minutes later, QL.ap failed to execute the second migration and instead continued to extend a dendritic protrusion with a growth cone to the posterior. **F)** 60 minutes after imaging, a QL.ap cell executed the second migration. **G)** At 120 minutes after imaging, the QL.ap failed to complete the third stage of migration. A posterior dendritic protrusion also emanated from QL.ap at this time. **H)** A schematic diagram illustrating failures at each of the three steps of migration. Migration 2 and 3 failure was accompanied by premature extension of a posterior dendritic protrusion.

**Table 4. Location of PQR migration failure in position 4.**

| Genotype[1] | Position 4 | | non-position 4[4] |
|---|---|---|---|
| | 1st paradigm[2] | 2nd paradigm[2] | |
| vab-8(ev411) | 30 | 22 | 48 |
| vab-8(e1017) | 48 | 26 | 26 |
| vab-8(lq153) | 40 | 30 | 30 |
| vab-8(lq156) | 15 | 41 | 44 |
| efn-4(bx80) | 0 | 70 | 30 |
| lin-17(e1456) | 64 | 9 | 27 |
| lin-17(n671) | 55 | 21 | 24 |
| lin-17(lq202) | 45 | 23 | 32 |
| mab-20(ev574) | 0 | 50 | 50 |
| efn-4(bx80); vab-8(ev411) | 41 | 51 | 8 |
| efn-4(bx80); vab-8(e1017) | 51 | 42 | 7 |
| efn-4(bx80); lin-17(e1456) | 10* | 75 | 15 |
| efn-4(bx80); lin-17(n671) | 53 | 40 | 7 |
| efn-4(bx80); lin-17(lq202) | 48 | 40 | 12 |
| vab-8(ev411); lin-17(e1456) | 52 | 20 | 28 |
| vab-8(ev411); lin-17(lq202) | 72* | 7 | 21 |
| vab-8(ev411); lin-17(n671) | 69* | 16 | 15 |
| efn-4(bx80); mab-20(ev574) | 0 | 88* | 12 |
| vab-8(ev411); mab-20(ev574) | 23 | 52* | 25 |
| mom-2(or77)M+ | 3 | 16 | 81 |
| egl-20(mu39)[3] | 14 | 15 | 71 |
| cwn-2(ok895) | 1 | 8 | 91 |
| cwn-1(ok546) | 0 | 0 | 100 |
| lin-44(n1792) | 0 | 0 | 100 |
| cwn-2(ok895); mom-2(or77)M+ | 2 | 31 | 67 |

[1] For each genotype, 100 animals were scored on three separate occasions for a total of 300 animals.

[2] Numbers are average percentages from three replicates of cells at each position rounded to the nearest whole number.

*$p < 0.05$

[3] 100 animals with PQR in position 4 or 5 were scored.

[4] "non-position 4" includes cells that migrated to the wild-type position 5 as well as cells that migrated anteriorly in positions 3, 2, and 1.

mutants. In *efn-4* mutants, which only displayed stage 3 failures, misplaced PQR were always immediately anterior to the anus (Fig 1E). In *vab-8* and *lin-17* mutants, misplaced PQR were sometimes at a more anterior location (Fig 1D and 1F). PQR neurons that failed to migrate posteriorly but resided in a more anterior position were designated paradigm 1, and those that were located immediately anterior to the anus were designated paradigm 2 (Table 4). *efn-4* mutants displayed all paradigm 2 (Table 4), consistent with failure of migration stage 3. *lin-17* and *vab-8* displayed both paradigms 1 and 2 (Table 4), consistent with failures at stages 1, 2, and, 3.

### The *wnts egl-20, mom-2*, and *cwn-2* were required for posterior PQR migration

As *lin-17* encodes a Frizzled Wnt receptor, PQR migration was scored in *Wnt* mutants. Previous studies showed that in *egl-20/Wnt* mutants, PQR migrated anteriorly due to its role in acutely inhibiting anterior migration of QL descendants and activating *mab-5* in the QL lineage [15]. *cwn-2/Wnt* acted downstream of *mab-5* along with *egl-20/Wnt to* acutely inhibit

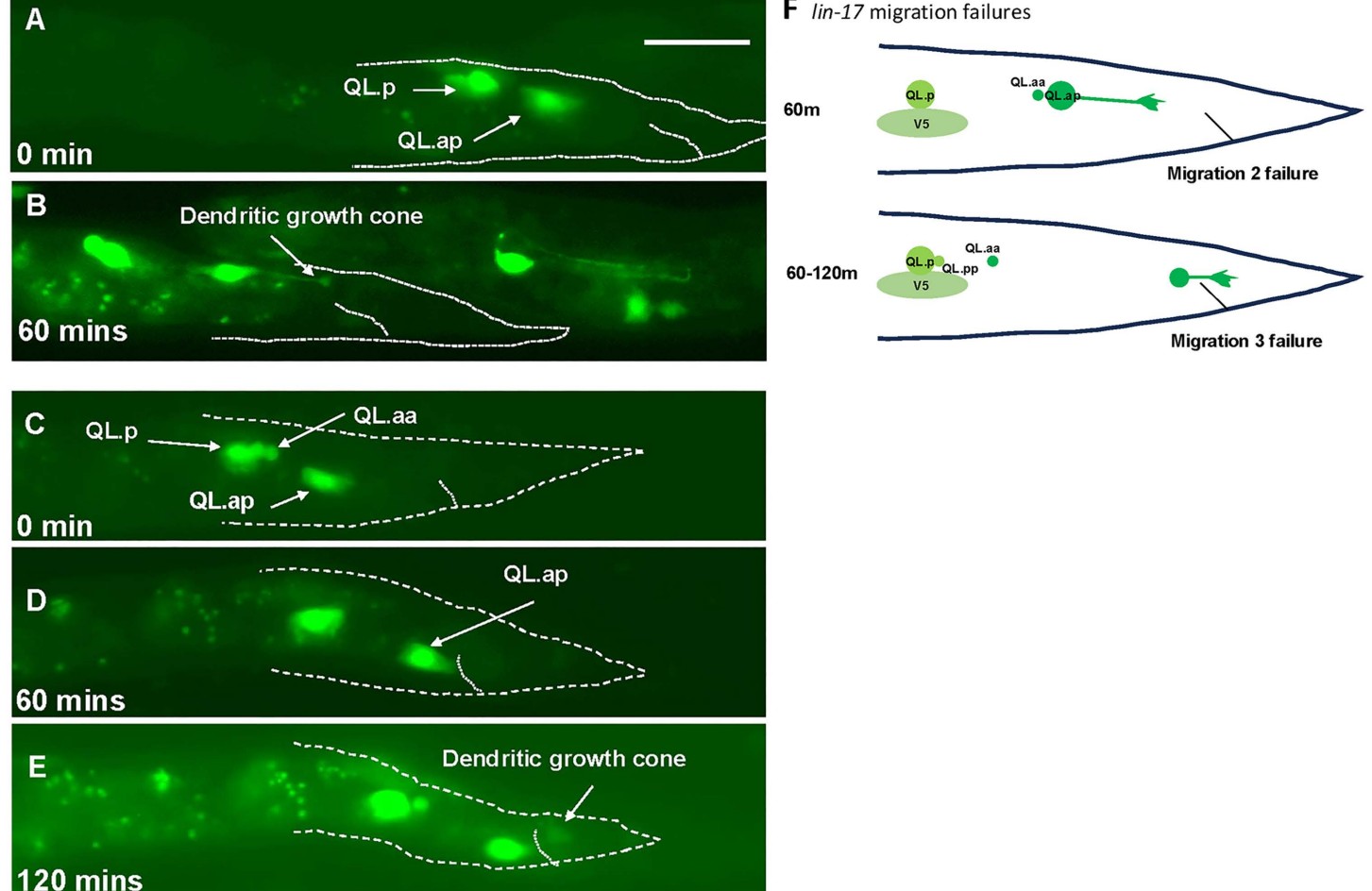

**Fig 5. Time-lapse imaging of QL.a and QL.ap migration in *lin-17(e1456)*.** Fluorescent micrographs of L1 *lin-17(e1456)* animals immobilized and imaged over time lapse as described in Fig 3. A and B are the same animal, and C-E are the same animal. The scale bar in A represents 10μm. **A)** At the start of imaging at 4-5 hours post-feeding, QL.ap completed the first stage of migration and had migrated posterior to QL.p. **B)** At 60 minutes after imaging began, QL.ap failed in the second stage of migration and prematurely extended a posterior dendritic protrusion. **C)** At the beginning of imaging 4-5 hous after feeding, QL.ap had completed the first stage of migration and resided posterior to QL.p. **D)** After 60 minutes, QL.ap completed the second stage of migration and resided immediately anterior to the anus. **E)** After 120 minutes, QL.ap failed in the third stage of migration and prematurely extended a posterior dendritic protrusion. **F)** A schematic summarizing the migration failures and premature dendritic protrusion in *lin-17(e1456)* mutants.

anterior QL descendant migration [15]. In the *egl-20(mu39)* hypomorph, 54% of PQRs migrated anteriorly (Table 2 positions 1–3), with the remaining 47% residing at the place of birth or at the normal PQR position in the tail (Table 2 positions 4 and 5). The position of PQRs that did not migrate anteriorly was scored relative to paradigms 1 and 2 (Table 4). *egl-20(mu39)* showed 14% migration failure at paradigm 1 and 15% at paradigm 2 (Table 4). *cwn-1* mutants alone displayed no defects. Thus, in addition to inhibiting anterior migration and activating *mab-5* expression, *egl-20* is also involved in posterior QL.ap migration.

*lin-44* was previously shown to repel posteriorly-migrating AQR and PQR in *egl-20 cwn-2; cwn-1; lin-44* quadruple mutants (Josephson *et al.* 2016). Alone, *lin-44* mutants displayed no defects in PQR migration (Tables 2 and 4). *mom-2* was previously shown to have weak defects in posterior PQR migration. This was confirmed, with *mom-2* mutants showing 3% paradigm 1 and 16% paradigm 2 PQR posterior migration defects (Tables 2 and 4). *cwn-2* mutants showed 1%

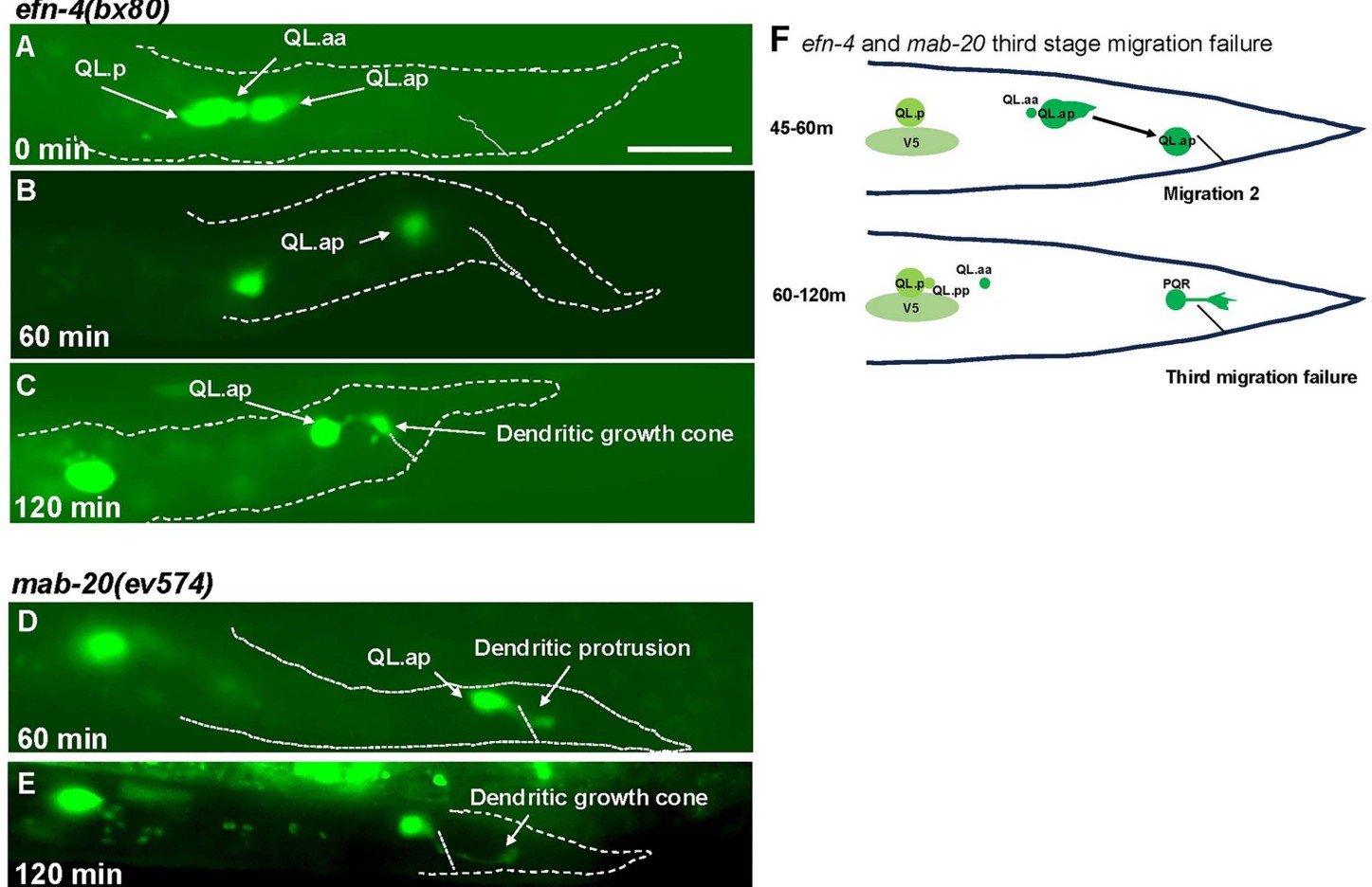

**Fig 6. Time-lapse imaging of QL.a and QL.ap migration in *efn-4(bx80)* and *mab-20(ev574)*.** Fluorescent micrographs of L1 animals immobilized and imaged over time lapse as described in Fig 3. A- C are the same *efn-4(bx80)* animal; and D and E are the same *mab-20(ev574)* animal. The scale bar in A represents 10µm. **A)** At 4-5 hours after feeding, QL.a in *efn-4(bx80)* had migrated posterior to QL.p and divided. **B)** After 60 minutes, QL.ap had completed the second stage of migration and resided immediately anterior to the anus. **C)** After 120 minutes, QL.ap had failed to complete the third migration and extended a premature posterior dendritic process. **D)** In a *mab-20(ev754)* animals 60 minutes after imaging, QL.ap had completed the second migration and resided immediately anterior to the anus. **E)** After 120 minutes, QL.ap failed to execute the third migration and prematurely extended a posterior dendritic process. **F)** A schematic diagram of the third migration failure in *efn-4* and *mab-20* mutants.

paradigm 1 and 8% paradigm 2 posterior migration defects. *cwn-2; mom-2* double mutants showed an additive increase in paradigm 2 defects (31%) (Table 4). These results indicate that the Wnts EGL-20, MOM-2, and CWN-2 are required to complete stages 2 and 3 of posterior PQR migration, consistent with the *lin-17* phenotype.

### *vab-8, lin-17,* and *efn-4* do not genetically synergize in PQR migration

*vab-8, lin-17,* and *efn-4* might define parallel, redundant pathways that regulate posterior migration. Genetic synergy, where the phenotype of double mutants is significantly stronger than the additive effects of the single mutants alone, is indicative of redundant, parallel pathways. However, double mutants of *vab-8, lin-17,* and *efn-4* showed no synergistic genetic interactions in PQR migration (Table 1). The number of PQRs at position 4 generally resembled the single mutants alone. While there were fewer cells at *wild-type* position 5 in double mutants, this could be an additive effect of cells that

migrated anteriorly in *vab-8* and *lin-17* single mutants. The lack of synergistic genetic interactions suggests that these genes do not define parallel, redundant pathways and instead might act together.

Double mutants of *efn-4* with *vab-8* and *lin-17* had failures in both paradigms 1 and 2, similar to *lin-17* and *vab-8* alone (Table 4). One exception was *lin-17(e1456); efn-4(bx80)*, which had significantly fewer cells in paradigm 1 compared to *lin-17(e1456)*. The nature of this interaction is not understood. These results suggest a stepwise migration process wherein earlier migration failures (paradigm 1) take precedence over later migration failures (paradigm 2). Double mutants of *vab-8* and *lin-17* resembled additive effects of single mutants and did not display genetic synergy.

In sum, these double mutant analyses showed no consistent evidence of genetic synergy and are consistent with *efn-4, vab-8,* and *lin-17* acting in the same or independent pathway.

### *mab-20*/Semaphorin acts in the Q lineage downstream of *mab-5* to control QL.ap migration

*efn-4* and *mab-20* are both known to interact in a variety of different developmental contexts, such as male tail morphogenesis and epiboly and ventral closure during embryogenesis [36,37,49,50]. AQR and PQR migration was scored in four *mab*-20 mutants. All four *mab-20* mutants displayed failure of PQR posterior migration, with misplaced cells residing in position 4 (Table 2 and Fig 1G). *ev574* and *bx24* showed 50% and 55% of PQR in position 4. *ev574* is predicted to be a null allele as it contains a deletion of 1479 bp, deleting the first exon [50]. *bx61* and *ev778* showed fewer defects (4% and 36%), suggesting that they might be hypomorphic. Misplaced PQR neurons resided immediately anterior to the anus, similar to *efn-4* mutants (paradigm 2, Table 4). Indeed, live time-lapse imaging of *mab-20(ev574)* revealed failure of the third stage of QL.ap migration and premature extension of a posterior dendritic process, similar to *efn-4* (Table 3 and Fig 6D–6E). These data indicate that *mab-20* is required for the third stage of QL.ap migration, similar to *efn-4* (Fig 6F).

*mab-20* cDNA expression driven by the *egl-17* promoter partially but significantly rescued *mab-20(ev574)* PQR migration defects (47% to 73% wild-type PQR position) (Table 1). While the rescue was partial, similar to *vab-8* and *efn-4* rescue, these data suggest that *mab-20* can act at least in part cell autonomously in the Q lineage to control QL.ap migration.

*mab-20* was expressed in the Q lineage [15], but expression was not significantly affected by *mab-5* lof or gof mutation (Table 1). However, *mab-20* significantly suppressed the posterior migration of AQR in *mab-5* gof backgrounds, with significantly more AQR in position 4 compared to *mab-5(e1751)* and *lqIs220* gof (Table 2 and Fig 2E). Furthermore, PQR migration in double mutants resembled *mab-20* mutants alone, with significantly more PQR neurons in position 4 compared to the *mab-5* gof alone (Table 2).

*mab-20; vab-8* and *mab-20; efn-4* double mutants showed additive but not synergistic defects in PQR migration and for the proportions of cells in both paradigms (Tables 2 and 4). For example, *vab-8(ev411)* showed 50% of PQR in position 4, *mab-20(ev574)* showed 50%, and *vab-8(ev411); mab-20(ev574)* showed 70%. Interestingly, only the second paradigm of migration defect was enhanced compared to *vab-8(ev411)* singles in *vab-8(ev411); mab-20(ev574),* suggesting a redundant role of *mab-20* and *vab-8* (Table 4).

Similarly, *efn-4(bx80);mab-20(ev574)* also showed an additive effect (Table 2). The double had a penetrance of 88% whereas *efn-4* and *mab-20* alone showed a penetrance of 71% and 50% respectively, with all cells showing the second paradigm of migration defect, suggesting a redundant role of *efn-4* and *mab-20* (Tables 2 and 4).

Together, these data suggest that MAB-20 acts with EFN-4, LIN-17, and VAB-8 act together to control the third stage of QL.ap migration.

### *efn-4* and *mab-20* can partially compensate for each other

The ability of *efn-4* transgenic expression to rescue *mab-20* mutants, and *vice versa* was tested. *mab-20(ev574)*; Pegl-17::efn-4 animals displayed 83% of PQR in wild-type position 5, compared to 47% in *mab-20(ev574)* alone (Table 2). *efn-4(bx80)*; Pegl-17::mab-20 animals displayed 64% wild-type position 5 compared to 29% in *efn-4(bx80)* alone (Table 2).

These data indicate that transgenic expression of one gene can partially compensate for loss of the other, consistent with them acting in a common pathway.

## Materials and methods

### Genetics

All experiments were carried out at 20°C using standard *C.elegans* techniques [51,52]. Mutations used in this study were as follows: LGI: *lin-17(e1456, n671, lq202), mab-20(ev574, bx24, bx61, ev778), lin-44(e1792).* LGII: *lqIs244 [Pgcy-32::cfp], ayIs9 [Pegl-17::gfp], cwn-1(ok546).* LGIV: *efn-4(bx80, e36), egl-20(mu39), cwn-2(ok895).* LGV: *vab-8(ev411), vab-8(e1017), mom-2(or77)/nT1.* LGX: *lqIs220 [Pegl-17::mab-5::gfp]. lqIs402, lqIs403, and lqIs404* were generated by integrating the *lqEx1353[Pegl-17::efn-4]* transgene*. lqIs413, lqIs414, lqIs415* were generated by integrating *lqEx1367[Pegl-17::mab-20]. lqIs416, lqIs417and lqIs418* were generated by integrating *lqEx1371[Pegl-17::vab-8L]. lqIs419* was generated by integrating *lqEx1376[Pegl-17::vab-8S].* Transgenes were integrated into the genome using exposure of L4 animals to short-wave ultraviolet radiation for 10 seconds, and subsequent screening of F2 and F3 generations for 100% transmission of the transgene [53]. Transgenes were outcrossed at least once to N2 before analysis. Wormbase was used for *C. elegans* informatics in the design, planning, and execution of experiments [41].

  *lin-17(lq202)* was isolated as a spontaneous mutant with PQR migration defects during an unrelated cross with the *lqIs244* transgene. The following sequence is deleted in *lin-17(lq202)*: TCTTCATCTCCTCCCTCCCATACCTCACGCCACTCTTCATCGATGCTCCGATCCGATCCTGCCACGCACTTGGAA.

  *vab-8(lq153)* and *vab-8(lq156)* were isolated in a forward genetic screen for new mutants with PQR migration defects. Whole-genome sequencing was used to identify the mutations, and both failed to complement *vab-8(e1017)* for PQR migration. *vab-8(lq153)* was a C to T mutation in affecting isoforms A and B and resulted in R127 to stop in *vab-8C. vab-8(lq156)* affected only the *vab-8A* long isoform and was a C to T mutation that resulted in Q46 to stop.

  *lq153:* AAGTTCATCTCGAGCATCTCC to AAGTTCATCTTGAGCATCTCC
  *lq156:* TCAACACTTGCAAATCGAAGG to TCAACACTTGTAAATCGAAGG

### Expression plasmid construction

Plasmids with cDNAs driven by the *egl-17* promoter were synthesized by Vectorbuilder (Chicago, IL USA). Sequences of the plasmids are included in Supporting Information (S1 File – S4 File). pEL1174: *Pegl-17::efn-4cDNA.* pEL1177: *Pegl-17::vab-8LcDNA.* pEL1178: *Pegl-17::vab-8ScDNA.* pEL1179: *Pegl-17::mab-20cDNA.*

### Scoring AQR and PQR position

To score the location of PQR and AQR, the *lqIs244[Pgcy-32::cfp]* transgene was used as a marker as described previously [15]: position 1 was the normal position of AQR in the anterior deirid in the head; position 2 was between the pharynx and the vulva; position three was around the vulva anterior to Q cell birthplace; position 4 was the Q cell birthplace; and position 5 was the normal position of PQR. For those cells that fail migration in position 4, paradigm 1 was near the QL.ap birthplace and paradigm 2 was immediately anterior to the anus. Significance of difference was determined by Fisher's exact test.

### L1 synchronization, immobilization, and time-lapse imaging

The *ayIs9[Pegl-17::gfp]* transgene was crossed into all the mutant strains and to observe Q lineage migration. Gravid hermaphrodites were washed from plates seeded with OP50 using M9 and bleached using standard protocols to collect embryos [15]. Eggs were allowed to hatch in M9 overnight without food at 20°C. Starved L1s were placed on a plate with OP50 *E.coli* and collected 4–5hr later for the imaging process. L1s were washed along with OP50 and loaded into into vivoVerse's vivoChip (Austin, TX USA) microfluidic chip operated by vivoVerse's automated microfluidic control system

(vivoCube+) following manufacturer instructions [54,55]. Immobilized animals were examined under a fluorescence compound microscope. This approach using the vivoChip microfluidic chip did not involve the usage of any paralytic, and the animals were able to freely feed on the *E. coli* that were washed off with the worms. While the L1 larvae could still move slightly in the microfluidic channel, the QL.a and QL.ap migration could be examined over the course of hours.

## Discussion

### MAB-5 controls three distinct stages in posterior QL.a and QL.ap migration

The Antennapedia-like Hox gene *mab-5* is a determinant of posterio Q neuroblast lineage posterior migration (Josephson *et al.* 2016). MAB-5 acts by first inhibiting anterior migration of QL descendants and then promoting posterior migration (Josephson *et al.* 2016). Both of these events likely require transcriptional changes driven by MAB-5 in the QL lineage. Q lineage FACS sorting and RNA seq in *mab-5* mutants revealed that expression of *vab-8, lin-17,* and *efn-4* is dependent on *mab-5* function. Work here demonstrates that these three genes mediate posterior migration downstream of *mab-5*: mutations in each cause failure in posterior migration of QL.a and/or QL.ap; mutations suppress posterior migration driven by *mab-5* ectopic expression; and *vab-8* and *efn-4* act at least in part cell-autonomously in the Q lineage.

Live time-lapse imaging of QL.a and QL.ap migration in *wild-type* and *vab-8, lin-17,* and *efn-4* mutants revealed three distinct stages of migration after division of QL to form QL.a and QL.p. First, QL.a migrates posteriorly over QL.p, which does not migrate, to reside to the posterior of QL.p. After QL.a divides to form QL.aa and QL.ap, the second stage involves posterior migration of QL.ap to a position immediately anterior to the anus. The third stage involves posterior migration of QL.ap posterior in the WT location, the final position, where differentiation into the PQR neurons begins, as evidenced by posterior extension of the dendritic process tipped with a growth cone. Each distinct migration involves the posterior extension of a lamellipodial protrusion, followed by posterior translocation of the cell body. This saltatory migration pattern is similar to the early migration of QL and QR. The third stage of QL.ap migration is particularly striking, wherein QL.ap extends a broad posterior lamellipodial protrusion posterior into the WT location, with the cell body undergoing somal translocation in the next hour. Strikingly, *vab-8, lin-17,* and *efn-4* affect distinct stages. *vab-8* is required for all three stages, *lin-17* is required for stages 2 and three, and *efn-4* is required only for the third and final stage of QL.ap migration.

### *vab-8* controls all three stages of migration downstream of *mab-5*

*vab-8* expression in the Q lineage showed a paired response to *mab-5*. Expression was reduced in *mab-5* lof and increased in *mab-5* gof. *vab-8* encodes a conserved, atypical kinesin-like KIF26 molecule with long and short isoforms [23,24]. The long isoform contains the kinesin motor domain. The short isoform is missing the motor domain. VAB-8 can bind to microtubules but lacks microtubule motor activity [56].

*vab-8(e1017)* introduces a premature stop codon that affects all isoforms. *vab-8(ev411)* is a 5' splice site mutation in the 3rd intron and is predicted to affect only the long isoform. This suggests that the long isoform is required for PQR migration. However, transgenic expression of the short isoform can partially but significantly rescue *vab-8(e1017)*, suggesting that the short isoform is also sufficient for PQR migration. These transgenic studies also indicate that *vab-8* acts at least in part cell-autonomously in the Q lineage to control PQR posterior migration.

*vab-8(ev411)* showed defects in all three stages of QL.a and QL.ap migration. In some cases, cells that displayed defects in stage 1 eventually reached their proper final destination. This suggests that the migration stages are independent and that deficits in migration in one stage can be accommodated in a later stage. Consistent with a role in all three stages, the final position of PQR varied in *vab-8*, from near the birth position (paradigm 1, reflective of stage 1 and 2 failures) to immediately anterior to the anus (paradigm 2, reflective of failure at the third stage of migration). *vab-8* suppressed ectopic *mab-5* gof in AQR and PQR, suggesting it acts downstream of *mab-5*. *vab-8* has been shown to control CAN cell and ALM mechanosensory neuron posterior migration [23,24]. *In vivo,* VAB-8 localizes to microtubule + ends at synapses, affects vesicle movement

along microtubules, and might regulate cargo delivery to synapses by pausing Dynein [56]. VAB-8 is also involved in gap junction formation and maintenance [57]. The VAB-8 long isoform is required for the cell surface localization of the SAX-3 receptor in the ALM neuron [27]. In QL.a migration, VAB-8 might be involved in the translocation of transmembrane or secreted molecules to the cell surface that drive posterior migration. Thus, MAB-5/Hox might drive posterior migration of QL.a by driving expression of VAB-8, which alters the cell surface interactions on QL.a such that posterior migration can occur.

### A novel role of *lin-17*/*Fz* downstream of *mab-5*

*lin-17* expression was reduced in one *mab-5* lof condition. *lin-17* encodes a Frizzled Wnt receptor that has been shown to act in canonical Wnt signaling to activate *mab-5* expression in QL [5–14]. Consistent with this, *lin-17* mutants showed anterior migration of PQR to the head near the normal position of AQR. *lin-17* mutants also showed failure in posterior PQR migration among PQR that did not migrate anteriorly. PQRs in paradigm 1 and paradigm 2 were observed, consistent with a role of *lin-17* in stages 2 and 3 as revealed by live time-lapse imaging. In *lin-17; mab-5 gof* double mutants, anterior PQR migration was rescued, suggesting that *lin-17* acts upstream of *mab-5* to activate it in the QL lineage. However, posterior AQR and PQR migration defects resembling *lin-17* mutants alone remained. This suggests that *lin-17* acts downstream of *mab-5* in posterior PQR migration. Thus, *lin-17* acts both upstream and downstream of *mab-5*. Possibly, *mab-5* is required for continued expression of *lin-17*/*Fz* in the QL lineage to drive posterior migration. The *Wnt* genes *egl-20, cwn-2,* and *mom-2* showed PQR migration defects, mostly paradigm 2. *mom-2; cwn-2* double mutants showed an additive increase suggesting that Wnts might act redundantly to drive posterior migration with *lin-17*. The involvement of Wnts in posterior QL lineage migration is consistent with their known role in regulating distinct aspects of anterior QR lineage migrations, which also occurs in three distinct stages each controlled by different Wnt ligands [58]. Thus, both anterior QR lineage migration and posterior QL lineage migration each occurs in three distinct stages controlled by Wnts, although in QL.ap the second and third stages are predominantly affected.

### *efn-4*/*Ephrin* controls the third stage of QL.ap migration downstream of *mab-5*

*efn-4* expression in the Q lineage was reduced in one *mab-5* lof condition. *efn-4* and *mab-5* act together in male tail morphogenesis, maintaining the repulsion between the Rn.a cells so the improper fusion of the male tail rays does not occur [59]. *efn-4* was necessary for posterior QL.ap migration. Final PQR position in paradigm 2 and live time lapse imaging showed that *efn-4* mutants failed in the third and final stage of QL.ap migration, resulting in PQR positioned immediately anterior to the anus. No defects were observed in stages 1 or 2. Transgenic expression and suppression experiments indicated that *efn-4* acts in the Q lineage downstream of *mab*-5 to mediate the third stage of QL.ap migration. Thus, *mab-5* might drive expression of *efn-4* in QL to drive posterior migration.

*efn-4* encodes an Ephrin-family secreted molecule attached to the plasma membrane via a lipid glycosyl-phosphatidylinositol anchor (a type A Ephrin). VAB-1 encodes the sole Ephrin receptor tyrosine kinase [35,48], a mutant of which had no effect on PQR migration (Table 2). Thus, EFN-4 likely acts through another pathway. EFN-4 was shown to interact functionally and physically with the LAD-2/L1CAM receptor and did not interact physically with VAB-1/EphR [60]. Thus, EFN-4 can act through non-EphR receptor pathways.

Results here suggest that EFN-4 can act cell-autonomously in the Q lineage. Ephrins are known to mediate both forward signaling (wherein they act as a ligand) and reverse signaling (wherein they act as a receptor) [61,62]. Possibly, EFN-4 reverse signaling mediates posterior QL.ap migration. Such signaling would likely require a co-receptor on the QL.ap cell, as EFN-4 has no transmembrane domain and is an extracellular molecule with a GPI anchor. Ephrins are classically understood as mediating repulsive responses, resulting in collapse of the actin cytoskeleton in migrating cells or growth cones [61,62], a role consistent with EFN-4 in male tail development, ensuring that ray cells do not ectopically fuse with one another [38]. Results here indicate that EFN-4 might be required for a migratory event. The cellular basis of EFN-4 function in QL.ap is not known, but it is necessary for the formation of the broad lamellipodial protrusion that

emanates posteriorly from QL.ap in the third stage of migration. It is also possible that EFN-4 prevents QL.ap interactions and adhesion with the surrounding environment, allowing the broad lamellipodial protrusion to form and posterior migration to occur, a role more akin to repulsion. Future studies will be aimed at identifying co-receptors that act with EFN-4 as well as the cellular basis of EFN-4 function in posterior QL.ap migration.

### *mab-20*/*Semaphorin* acts in the third stage of QL.ap migration

In male tail ray cell fusion, *mab-20* acts through *efn-4* to prevent ectopic ray cell fusion [59]. MAB-20 belongs to the Semaphorin class of guidance cues [50] that, similar to Ephrins, are known for repulsive activity and actin cytoskeleton collapse [63]. *mab-20* mutants displayed PQR migration defects and QL.ap third stage migration failure similar to *efn-4. efn-4; mab-20* double mutants had additive effects, suggesting some distinct function of the molecules but consistent with them acting together. Indeed, transgenic expression of *mab-20* partially rescued *efn-4* and vice versa, to levels similar to rescue with the corresponding transgenes. This is consistent with EFN-4 and MAB-20 acting in a common pathway. While *mab-20* expression is enriched in the Q lineage, its expression is not significantly affected by *mab-5.* Thus, *mab-20* is likely not regulated by *mab-5.* However, *mab-5* could likely encode a constitutively-expressed cofactor necessary for EFN-4 function in posterior migration. In other words, EFN-4 might be the impetus for posterior migration downstream of *mab-5* but requires *mab-20* to execute the program. EFN-4 and MAB-20 are both molecules with known repulsive effects, suggesting that similar repulsive mechanisms are involved in posterior QL.ap migration. They might mediate migration away from an anterior signal, or perhaps prevent adhesion of QL.ap with surrounding cell, allowing posterior migration to occur.

### Failure of migration triggers premature dendritic extension

One of the striking findings of this study is that failure of migration is accompanied by premature extension of a dendritic protrusion, indicative of premature neuronal differentiation or terminal differentiation. For example, failures of the second stage of migration in *vab-8* and *lin-17* mutants resulted in immediate extension of a posterior dendrite, approximately two hours earlier than wild-type. Failure at the third stage of migration in *vab-8, lin-17, efn-4,* and *mab-20* resulted in a dendritic protrusion approximately one to two hours earlier than wild type. This certainly suggests that the migratory program involving these genes was responsible for not only promoting posterior migration, but also inhibiting extension of a dendrite and possible premature neuronal differentiation. This suggests that there is a continuum between migration and dendritic extension. Possibly, when genes facilitating migration are present, they facilitate the extension of a migratory lamellipodium, which leads to migration. In the absence of these genes, the cell instead sends out a dendrite. EFN-4 activity and interactions are key to this developmental fulcrum. Given the timing of dendrite extension, it is unlikely a transcriptional program is involved. More likely is that EFN-4 interactions with the extracellular matrix or cell surface dictate a migratory versus dendrite developmental decision.

In *vab-8* mutants with first stage failures, in other words a failure of QL.a migration, premature dendrite extensions were not observed. After the first migration, QL.a divides to form QL.aa and QL.ap, the latter QL.ap becoming PQR. This suggests that QL.a division must occur before QL.ap can extend a premature dendrite. As QL.ap will become PQR, this suggests that QL.ap has already begun to differentiate into PQR before reaching its final position, and migration failure results in premature dendritic extension.

### A transcriptional program driving posterior QL.a migration downstream of MAB-5

Results here show that *vab-8, lin-17,* and *efn-4* are regulated by MAB-5 in the Q lineage that are required for MAB-5 to drive posterior QL.a and QL.ap migration. VAB-8 is a conserved, atypical Kinesin involved in vesicle movement and cell surface localization of the SAX-3 receptor [27,56]. It is possible that VAB-8 mediates the translocation of distinct secreted or transmembrane molecules to the plasma membrane that mediate the distinct steps of QL.a and QL.ap migration (Fig 7). An as-yet unidentified molecule might be localized by VAB-8 to mediate the first migration of QL.a. This molecule could itself

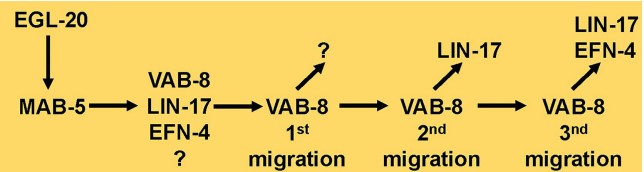

**Fig 7. A potential model of a transcriptional program downstream of MAB-5 in posterior QL.a and QL.ap migration.** MAB-5 expression in the QL lineage is activated by EGL-20/Wnt and canonical Wnt signaling. MAB-5 drives the expression of VAB-8/KIF26, LIN-17/Fz, EFN-4/Ephrin, and other unidentified factors in QL. After QL division, VAB-8 might localize an unidentified factor to the plasma membrane to execute the first stage of QL.a posterior migration. After QL.a division, VAB-8 might localize LIN-17 to the plasma membrane of QL.ap to execute the second stage of QL.ap migration. VAB-8 might then localize EFN-4 and LIN-17 to the plasma membrane of QL.ap to execute the third and final stage of QL.ap migration.

be regulated by MAB-5 or be a constitutively-expressed factor that requires VAB-8 to reach the plasma membrane. Translocation of LIN-17 by VAB-8 might trigger the second migration, and translocation of LIN-17 and EFN-4 might trigger the third stage. In any case, this work utilized Q cell sorting and RNA seq in *mab-5* mutant backgrounds to identify potential genes regulated by MAB-5 and shows that VAB-8, LIN-17, and EFN-4 define a novel transcriptional program downstream of MAB-5 to drive posterior neuroblast migration. This work links cell surface molecule expression to a canonical Wnt signaling pathway involving EGL-20 mediated by a Hox factor. As a Hox terminal selector factor, MAB-5 might subtly alter the interaction of QL.a and QL.ap with the extracellular environment to achieve posterior migration without affecting deeper levels of neuron fate determination or differentiation.

## Supporting information

**S1 File. The sequence of pEL1174 (*PegI-17::efn-4cDNA*).**
(GB)

**S2 File. The sequence of pEL1177 (*PegI-17::vab-8LcDNA*).**
(GB)

**S3 File. The sequence of pEL1178 (*PegI-17::vab-8ScDNA*).**
(GB)

**S4 File. The sequence of pEL1179 (*PegI-17::mab-20cDNA*).**
(GB)

## Acknowledgments

The authors thank members of the Lundquist and Ackley research groups for helpful discussion, and Wormbase for *C. elegans* informatics. Some strains were provided by the Caenorhabditis Genetics Center (NIH Office of Research Infrastructure Programs (P40OD101440)).

## Author contributions

**Conceptualization:** Vedant D. Jain, Erik A. Lundquist.

**Data curation:** Vedant D. Jain, Erik A. Lundquist.

**Formal analysis:** Vedant D. Jain, Erik A. Lundquist.

**Funding acquisition:** Erik A. Lundquist.

**Investigation:** Vedant D. Jain.

**Methodology:** Vedant D. Jain.

**Project administration:** Erik A. Lundquist.

**Supervision:** Erik A. Lundquist.

**Validation:** Vedant D. Jain.

**Visualization:** Vedant D. Jain.

**Writing – original draft:** Vedant D. Jain.

**Writing – review & editing:** Vedant D. Jain, Erik A. Lundquist.

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
