## [Decision Letter · Decision Letter 0]

18 Jul 2025

PGENETICS-D-25-00704

VAB-8/KIF26, LIN-17/Frizzled, and EFN-4/Ephrin, control distinct stages of posterior neuroblast migration downstream of the MAB-5/Hox transcription factor in Caenorhabditis elegans

PLOS Genetics

Dear Dr. Lundquist,

Thank you for submitting your manuscript to PLOS Genetics. After careful consideration, we feel that it has merit but does not fully meet PLOS Genetics's publication criteria as it currently stands. Therefore, we invite you to submit a revised version of the manuscript that addresses the points raised during the review process.

Please submit your revised manuscript within 60 days Sep 16 2025 11:59PM. If you will need more time than this to complete your revisions, please reply to this message or contact the journal office at plosgenetics@plos.org. Please include the following items when submitting your revised manuscript:

We look forward to receiving your revised manuscript.

Kind regards,

Laura Bianchi

Academic Editor

PLOS Genetics

Pablo Wappner

Section Editor

PLOS Genetics

Aimée Dudley

Editor-in-Chief

PLOS Genetics

Anne Goriely

Editor-in-Chief

PLOS Genetics

**Journal Requirements:**

https://journals.plos.org/plosgenetics/s/submission-guidelines#loc-parts-of-a-submission

- ® on page: 20.

5) We have noticed that you have uploaded Supporting Information files, but you have not included a list of legends. Please add a full list of legends for your Supporting Information files after the references list.

6) We are unable to open the following Supporting Information file: Supplemental Files.zip. Please kindly revise as necessary and re-upload.

7) Please amend your detailed Financial Disclosure statement. This is published with the article. It must therefore be completed in full sentences and contain the exact wording you wish to be published.

**Reviewers' comments:**

Reviewer's Responses to Questions

**Comments to the Authors:**

Reviewer #1: This manuscript investigates how the Hox gene mab-5 regulates posterior neuroblast migration in C. elegans via three candidate downstream effectors: vab-8/KIF26, lin-17/Frizzled, and efn-4/Ephrin. Using RNA-seq of Q-lineage cells, live time-lapse imaging, genetic mutants, and rescue constructs, the authors propose that these genes function at distinct, sequential stages of QL.ap migration. The main strength of the study is the methodological phenotypic characterization of the three mutants, as well as the rigorous assessment of several alleles for each gene. However, several gaps in the analysis remain, which would require significant revision before publication.

Major Points:

The data do not strongly support that the three genes are mab-5 targets. The changes reported in RNA-seq are not very strong or consistent across the lof and gof alleles. In general, RNA-seq data need to be validated by methods such as transcriptional reporters or qRT-PCR.

Related to the previous point: could the authors analyze (at least in silico) the promoter regions of lin-17, vab-8, and efn-4 to test whether they contain mab-5 binding sites?

The conclusion about cell autonomy could be strengthened given that the rescue was weak. How many lines were tested? Have the authors compared their rescue to one in which vab-8 and efn-4 are expressed under their own promoters or a promoter that is more broadly expressed?

I was unable to understand the interpretation of some of the double mutants. Particularly in the case of lin-17 and vab-8, the effect seems additive and would support them functioning in separate pathways. In any case, the authors’ conclusion that “these genes act in the same pathway or in independent pathways” is not informative because it proposes two interpretations that are the opposite of each other.

The authors propose that VAB-8/KIF26 functions to traffic another molecule (abstract and discussion). This conflicts with several papers from mammals and worms that show that VAB-8/KIF26 is immotile. They should suggest a different model, one that is consistent with the literature.

Minor Points:

The manuscript is long and hard to read. It would benefit from shortening and editing.

In the abstract: “Surpisingly” – correct spelling.

Consider adding a diagram of the proteins and the mutations for readability.

It was unclear to me how the data support a role for lin-17 both upstream and downstream of mab-5. This is not a major point, but if the authors have reporters that could show them affecting each other’s levels, that would be helpful.

Reviewer #2: In this manuscript, the authors characterize the role of three genes, vab-8/kif26, lin-17/Fz and efn-4/Ephrin, in migration of a Q neuroblast (QL) and its descendants in C. elegans. These genes were identified in a previous study that used RNAseq on FACS-sorted Q cells to identify differentially regulated genes in mab-5/Hox gain and loss of function mutants. The authors further contribute to our understanding of QL migration through time-lapse studies showing that QL descendants reach their final positions in the tail through three distinct stages, each characterized by a pause and lamellipodia formation. They demonstrate that vab-8, lin-17, and efn-4 mutants exhibit PQR migration defects, with the interesting finding that they act at distinct stages of QL descendant migration. The authors show that these genes act cell autonomously and conduct detailed genetic analyses using double mutants and mab-5 gain-of-function suppression experiments, with results consistent with their function as downstream targets of MAB-5.

Overall, this manuscript is mostly well written (see comments below) and presents some novel insight into cell migration using the well-established Q neuroblast migration model. In particular, they characterize, what I believe are the first downstream targets of MAB-5/HOX in QL cells, VAB-8, LIN-17, and EFN-4 and show that distinct transcriptional programs are activated at specific stages during the long migration of Q descendants from the mid-body to the tail.

I find the methodology to be sound and the results convincing. However, I feel that the manuscript could have gone a bit further to describe what one or more of these genes are doing to promote QL migration at one or more of the three stages. The Discussion section raises the possibility that VAB-8 may regulate the trafficking of LIN-17 and/or EFN-4 to the membrane, or that EFN-4 may play a role in promoting lamellipodial protrusions. It seems like an omission not to test some of these ideas in this study. For example, the manuscript would be greatly strengthened by including one or more of the studies outlined below. That said, it may also be reasonable to argue that these experiments are not essential to support the main conclusions or are not easily performed due to confounding factors.

1. While I find the data that these genes are expressed in QL cells strong and convincing, I was surprised that the RNA-seq data was not validated through the most direct means of determining if the expression of vab-8, lin-17, and efn-4 from either GFP (or some other fluorophore) transcriptional/translation reporters or endogenously tagged genes is downregulated or off in mab-5 mutants. For example, this could be performed with reporters such as wyEx450(Plin-17::lin-17::GFP) (Klassen and Shen, 2007), juIs109 (EFN-4::GFP) (Chin-Sang et al., 2002), and vab-8p::VAB-8::GFP (Wolf et al, 1998). It would also be interesting to see if these genes are expressed dynamically during QL migration given the saltatory nature of the migration. For example, is LIN-17 present early to initiate QL posterior migration through the wnt canonical pathway, but then downregulated and later upregulated at the second and third stages of migration. In the discussion, it was suggested that mab-5 may be acting to maintain lin-17 expression. This might be testable by examining the expression from wyEx450(Plin-17::lin-17::GFP) or a similar reporter in a mab-5 mutant background.

2. A polarized (oriented toward the posterior) lamellipodia forms at each stage of migration. The time lapse studies (using one of the available Q cell membrane markers) could be strengthened by also showing how vab-8, lin-17, and/or efn-4 mutants affect lamellipodial dynamics or polarity at each stage. For example, do lamellipodia polarize normally but fail to initiate migration in a timely manner. Are there differential effects? In the 3rd stage, the cell body undergoes somal translocation. Is this process differentially affected in the three mutants?

3. The model presented in Fig 7 suggests that VAB-8 may be involved in trafficking LIN-17 and EFN-4 to the plasma membrane to promote migration. This model would be greatly strengthened if there was evidence that LIN-17 and/or EFN-4 were mislocalized in vab-8 mutants. These studies may be possible with some of the reporters mentioned in comment #1.

There are also a few minor Issues:

1. The manuscript contained numerous typos and possible sentence structure errors, some of which I have listed below for clarity (it would be helpful to include page and line numbers).

A. Abstract – why say First, if there is no Second of Third point…

B. Intro – paragraph 1 - remove word ‘is’

C. Paragraph 2, egl10/wnt – should this be egl-20.

D. ‘Paradigm’ is mentioned in Fig 2B but not defined in the results section until much later in the results. When reading figure legend, did not know what paradigm referred to.

E. mab-20/Semaphorin, which acts with efn-4 n prevention of male tail

F. Figure 1 legend: ….. different genotypes using the lqIs244[Pgcy-32::gfp] -32::gfp]

2. I’m not sure if double mutant is the correct term to describe a single mutant in a transgenic background, for example: In vab-8(ev411); Pegl-17::mab-5 and vab-8(e1017); Pegl-17::mab-5 double mutants….

3. The first section of the results dealing with Table 1 is probably not necessary as these are findings from the previously published study and were already mentioned and cited in the introduction.

4. I believe one of the studies by Wightman et al. 1996 or Wolf et al. 1998 showed that vab-8 mutants had PQR migration defects. This should probably be mentioned again in the results section when the vab-8 PQR migration defect is described.

5. In Table 3, I wasn’t quite sure at first what was meant by ‘1 and WT’. It took me a few seconds to get that it was normal PQR position posterior to the anus. Maybe add: 1 WT = normal PQR position posterior to the anus or something like that at the bottom of the Table.

Reviewer #3: The manuscript review has been uploaded as an attachment.

**Have all data underlying the figures and results presented in the manuscript been provided?**

Reviewer #1: Yes

Reviewer #2: Yes

Reviewer #3: Yes

PLOS authors have the option to publish the peer review history of their article (what does this mean? ). If published, this will include your full peer review and any attached files.

**Do you want your identity to be public for this peer review?** For information about this choice, including consent withdrawal, please see our Privacy Policy .

Reviewer #1: No

Reviewer #2: No

Reviewer #3: No

**Figure resubmission:**
---

## [Decision Letter · Decision Letter 1]

18 Sep 2025

PGENETICS-D-25-00704R1

VAB-8/KIF26, LIN-17/Frizzled, and EFN-4/Ephrin, control distinct stages of posterior neuroblast migration downstream of the MAB-5/Hox transcription factor in Caenorhabditis elegans

PLOS Genetics

Dear Dr. Lundquist,

Thank you for submitting your manuscript to PLOS Genetics. After careful consideration, we feel that it has merit but does not fully meet PLOS Genetics's publication criteria as it currently stands. Therefore, we invite you to submit a revised version of the manuscript that addresses the points raised during the review process.

Please submit your revised manuscript within 60 days Nov 17 2025 11:59PM. If you will need more time than this to complete your revisions, please reply to this message or contact the journal office at plosgenetics@plos.org. Please include the following items when submitting your revised manuscript:

We look forward to receiving your revised manuscript.

Kind regards,

Laura Bianchi

Academic Editor

PLOS Genetics

Pablo Wappner

Section Editor

PLOS Genetics

Aimée Dudley

Editor-in-Chief

PLOS Genetics

Anne Goriely

Editor-in-Chief

PLOS Genetics

**Reviewers' comments:**

Reviewer's Responses to Questions

**Comments to the Authors:**

Reviewer #1: The revised manuscript does not adequately address the major concerns raised by this reviewer and by reviewer #2. Aside from incorporating publicly available ChIP-Seq data, the authors have not performed additional experiments and have not been responsive to requests to temper or adjust their claims.

In my view, it is not justified to describe lin-17, vab-8, and efn-4 as “MAB-5 targets” based solely on relatively weak RNA-seq data and their genetic requirement for a mab-5 gain-of-function phenotype. The authors offer several possible explanations for their inability to detect changes with the reporter constructs suggested by reviewer #2, but the most straightforward explanation is that these genes are not directly regulated by MAB-5. This concern is particularly relevant for efn-4, for which no binding sites are present.

Other important issues also remain unresolved, such as the interpretation of the double mutant phenotypes and the weak cell-autonomous rescue. I do not agree with the explanations the authors provide. For instance, their suggestion that the heterologous promoter may not drive expression at the correct time is inconsistent with their own imaging data, which clearly shows robust expression at the appropriate place and stage.

Finally, the authors remain unresponsive even to minor, easily correctable points. Examples include their decision to retain the RNA-seq table despite reviewer #2’s request for its removal, and the continued claim in the abstract that “VAB-8 traffics distinct molecules to the plasma membrane,” despite my previous comment that VAB-8, as an immotile motor, cannot serve such a trafficking role.

Reviewer #2: I have reviewed the revised manuscript and find that my concerns have been addressed.

Reviewer #3: The authors have appropriately addressed and corrected essentially all of the minor revisions flagged in the first version. This is now a well written and significant body of work, suitable for publication in PLoS Genetics.

**Have all data underlying the figures and results presented in the manuscript been provided?**

Reviewer #1: Yes

Reviewer #2: Yes

Reviewer #3: Yes

PLOS authors have the option to publish the peer review history of their article (what does this mean? ). If published, this will include your full peer review and any attached files.

**Do you want your identity to be public for this peer review?** For information about this choice, including consent withdrawal, please see our Privacy Policy .

Reviewer #1: No

Reviewer #2: No

Reviewer #3: No

**Figure resubmission:**
---

## [Editor Report · Decision Letter 2]

2 Oct 2025

Dear Dr Lundquist,

We are pleased to inform you that your manuscript entitled "VAB-8/KIF26, LIN-17/Frizzled, and EFN-4/Ephrin, control distinct stages of posterior neuroblast migration downstream of the MAB-5/Hox transcription factor in Caenorhabditis elegans" has been editorially accepted for publication in PLOS Genetics. Congratulations!

Yours sincerely,

Laura Bianchi

Academic Editor

PLOS Genetics

Pablo Wappner

Section Editor

PLOS Genetics

Aimée Dudley

Editor-in-Chief

PLOS Genetics

Anne Goriely

Editor-in-Chief

PLOS Genetics

BlueSky: @plos.bsky.social

Comments from the reviewers (if applicable):

**Data Deposition**

http://datadryad.org/submit?journalID=pgenetics&manu=PGENETICS-D-25-00704R2

**Press Queries**

---

## [Editor Report · Acceptance letter]

PGENETICS-D-25-00704R2

VAB-8/KIF26, LIN-17/Frizzled, and EFN-4/Ephrin, control distinct stages of posterior neuroblast migration downstream of the MAB-5/Hox transcription factor in Caenorhabditis elegans

Dear Dr Lundquist,

We are pleased to inform you that your manuscript entitled "VAB-8/KIF26, LIN-17/Frizzled, and EFN-4/Ephrin, control distinct stages of posterior neuroblast migration downstream of the MAB-5/Hox transcription factor in Caenorhabditis elegans" has been formally accepted for publication in PLOS Genetics! Your manuscript is now with our production department and you will be notified of the publication date in due course.

With kind regards,

Zsofia Freund

PLOS Genetics

On behalf of:
